# Distributed Momentum for Byzantine-resilient Stochastic Gradient Descent

**El-Mahdi El-Mhamdi**\*
École Polytechnique, France
`el-mahdi.el-mhamdi@polytechnique.edu`

**Rachid Guerraoui**\*
École Polytechnique Fédérale de Lausanne (EPFL), Switzerland
`rachid.guerraoui@epfl.ch`

**Sébastien Rouault**\*
École Polytechnique Fédérale de Lausanne (EPFL), Switzerland
`sebastien.rouault@epfl.ch`

## Abstract

Byzantine-resilient Stochastic Gradient Descent (SGD) aims at shielding model training from *Byzantine faults*, be they ill-labeled training datapoints, exploited software/hardware vulnerabilities, or malicious worker nodes in a distributed setting. Two recent attacks have been challenging state-of-the-art defenses though, often successfully precluding the model from even fitting the training set. The main identified weakness in current defenses is their requirement of a sufficiently low *variance-norm* ratio for the stochastic gradients. We propose a practical method which, despite increasing the variance, reduces the *variance-norm* ratio, mitigating the identified weakness. We assess the effectiveness of our method over 736 different training configurations, comprising the 2 state-of-the-art attacks and 6 defenses. For confidence and reproducibility purposes, each configuration is run 5 times with specified seeds (1 to 5), totalling 3680 runs. In our experiments, when the attack is effective enough to decrease the highest observed top-1 cross-accuracy by at least 20% compared to the *unattacked* run, our technique systematically increases back the highest observed accuracy, and is able to recover at least 20% in more than 60% of the cases.

## 1 Introduction

Stochastic Gradient Descent (SGD) is one of the main optimization algorithm used throughout machine learning. Scaling SGD can mean aggregating more but inevitably less well-sanitized data, and distributing the training over several machines, making SGD even more vulnerable to *Byzantine faults*: corrupted/malicious training datapoints, software vulnerabilities, etc. Many *Byzantine-resilient* techniques have been proposed to keep SGD safer from these faults, e.g. Alistarh et al. (2018); Damaskinos et al. (2018); Yang & Bajwa (2019b); TianXiang et al. (2019); Bernstein et al. (2019); Yang & Bajwa (2019a); Yang et al. (2019); Rajput et al. (2019); Muñoz-González et al. (2019). These techniques mainly use the same adversarial model (Figure 2): a central, trusted parameter server distributing gradient computations to several workers, a minority of which is controlled by an adversary and can submit arbitrary gradients.

Two families of defense techniques can be distinguished. The first employs *redundancy* schemes, inspired by coding theory. This approach has strong resilience guarantees, but

---

\*Author list written in alphabetical order, as for all the papers from the DCL at EPFL.

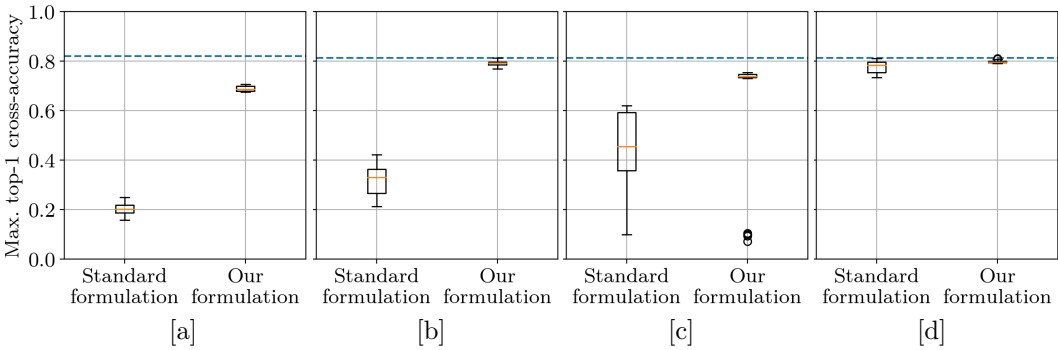

Figure 1: We report on the highest measured top-1 cross-accuracy while training under either of the two studied, state-of-the-art attacks. [a, b]: a convolutional model (Section 4.1) for CIFAR-10 under the attack from Baruch et al. (2019), and [c, d]: a fully connected model for Fashion-MNIST (Xiao et al., 2017) under the attack from Xie et al. (2019a). Roughly half the workers implements the attack in [a, c], and a quarter does in [b, d]; see Section 4.1. Each experiment is run 5 times. The dotted blue line is the median of the maximum top-1 cross-accuracy of the 5 runs without attack, and the boxes aggregate the maximum top-1 cross-accuracy obtained under attack with each 5 runs of the 6 studied defenses. Over 736 different combinations of attacks, defenses, datasets, etc (totalling of 3680 runs), our method consistently obtain at least similar, if not substantially *better performances* (lower minimal loss, higher maximal top-1 cross-accuracy) than the standard formulation. Notably, our formulation obtains these results with no additional computational complexity.

its requirement to share data between workers makes this approach unsuitable for several classes of applications, e.g. when data cannot be shared for privacy, scalability or legal reasons. The second family uses *statistically-robust* aggregation schemes, and is the focus of this paper. The underlying idea is simple. At each training step, the server aggregates the stochastic gradients computed by the workers into one gradient, using a function called a Byzantine-resilient *Gradient Aggregation Rule* (GAR). These statistically-robust GARs are designed to produce at each step a gradient that is expected to decrease the loss.

Intuitively, one can think of this second family as different formulations of the multivariate median. In particular, if the non-Byzantine gradients were all equal at each step, any different (adversarial) gradient would be rejected by each of these medians, and no attack would succeed. But due to their stochastic nature, the non-Byzantine gradients are different: their *variance* is strictly positive. Formal guarantees on any given statistically-robust GAR typically require that the *variance-norm* ratio, the ratio between the *variance* of the non-Byzantine gradients and the *norm* of the *expected* non-Byzantine gradient, remains below a certain constant (constant which depends on the GAR itself and fixed hyperparameters). Intuitively, this notion of *variance-norm* ratio can be comprehended quite analogously to the inverse of the *signal-to-noise* ratio (i.e. the *"noise-to-signal"* ratio) in signal processing.

However, Baruch et al. (2019) noted that an attack could send gradients that are close to non-Byzantine *outlier* gradients, building an apparent majority of gradients that could be sufficiently far from the expected non-Byzantine gradient to *increase* the loss. This can happen against most statistically-robust GARs in practice, as the *variance-norm* ratio is often *too* large for them. Two recent attacks (Baruch et al., 2019; Xie et al., 2019a) were able to exploit this fact to substantially hamper the training process (which our experiments confirm).

The work presented here aims at (substantially) improving the resilience of statistically robust GARs "also in practice", by reducing the *variance-norm* ratio of the gradients received by the server. We do that by taking advantage of an old technique normally used for acceleration: momentum. This technique is regularly applied at the server, but instead we propose to confer it upon each distributed worker, effectively making the Byzantine-resilient GAR aggregate *accumulated* gradients. Crucially, there is no computational complexity attached to our reformulation: it only *reorders* operations in existing (distributed) algorithms.

**Contributions.** Our main contributions can be summarized as follows:

- A reformulation of classical/Nesterov *momentum* which can significantly improve the effectiveness (Figure 1) of any statistically-robust *Gradient Aggregation Rule* (GAR). We formally analyze the impact of our reformulation on the *variance-norm* ratio of the aggregated gradients, ratio on which the studied GARs assume an upper bound.
- An extensive and reproducible[1] set of experiments substantiating the effectiveness of our reformulation of momentum in improving existing defenses against state-of-the-art attacks.

**Paper Organization.** Section 2 provides the necessary background. Section 3 presents our distributed momentum scheme and provides some intuitions on its effects. Formal developments of these intuitions are given in the appendix. Section 4 describes our experimental settings in details, before presenting and analysing some of our experimental results. The appendix reports on the entirety of our experiments, and details how they can be reproduced (in one command, graphs included). Section 5 discusses related and future work.

## 2 BACKGROUND

### 2.1 BYZANTINE DISTRIBUTED SGD

**Stochastic Gradient Descent (SGD).** We consider the classical problem of optimizing a non-convex, differentiable loss function $Q : \mathbb{R}^d \to \mathbb{R}$, where $Q(\theta_t) \triangleq \mathbb{E}_{x \sim \mathcal{D}}[q(\theta_t, x)]$ for a fixed data distribution $\mathcal{D}$. Ideally, we seek $\theta^*$ such that $\theta^* = \arg\min_\theta (Q(\theta))$.

We employ *mini-batch SGD* optimization. Starting from initial parameter $\theta_0 \in \mathbb{R}^d$, at every step $t \geq 0$, $b$ *samples* $\left(x_t^{(1)} \ldots x_t^{(b)}\right)$ are sampled from $\mathcal{D}$ to estimate one *stochastic gradient* $g_t \triangleq \frac{1}{b} \sum_{k=1}^b \nabla q\left(\theta_t, x_t^{(k)}\right) \approx \nabla Q(\theta_t)$. This stochastic gradient is then used to update the *parameters* $\theta_t$, with: $\theta_{t+1} = \theta_t - \alpha_t g_t$. The sequence $\alpha_t > 0$ is called the *learning rate*.

**Classical and Nesterov momentum** One field-tested amendment to mini-batch SGD is *classical momentum* (Polyak, 1964), where each gradient keeps an exponentially-decreasing effect on every subsequent update. Formally: $\theta_{t+1} = \theta_t - \alpha_t \sum_{u=0}^t \mu^{t-u} g_u$, with $0 < \mu < 1$.

Nesterov (1983) proposed another revision. Noting $v_t$ the *velocity vector*, $v_0 = 0$, formally:

$$v_{t+1} = \mu\, v_t + \frac{1}{b} \sum_{k=1}^b \nabla q\left(\theta_t - \alpha_t\, \mu\, v_t, x_t^{(k)}\right)$$

$$\theta_{t+1} = \theta_t - \alpha_t\, v_{t+1}$$

Compared to classical momentum, the gradient is estimated at $\theta_t - \alpha_t\, \mu\, v_t$ instead of $\theta_t$.

**Distributed SGD with Byzantine workers.** We follow the parameter server model (Li et al., 2014): one single process (the *parameter server*) holding the parameter vector $\theta_t \in \mathbb{R}^d$, and $n$ other (the *workers*) estimating gradients. Among these $n$ workers, up to $f < n$ are said *Byzantine*, i.e. adversarial. Unlike the other $n - f$ *honest* workers, these $f$ *Byzantine* workers can submit arbitrary gradients (Figure 2).

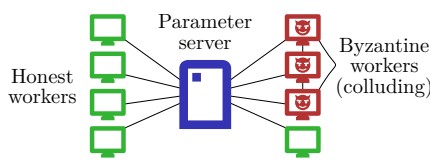

At each step $t$, the parameter server receives $n$ different gradients $g_t^{(1)} \ldots g_t^{(n)}$, among which $f$ are arbitrary (submitted by the Byzantine workers). So the update equation becomes: $\theta_{t+1} = \theta_t - \alpha_t\, G_t$, where:

Figure 2: A parameter server setup with $n = 8$ workers, among which $f = 3$ are *Byzantine* (i.e., adversarial) workers. A black line represents a bidirectional communication channel.

$$G_t \triangleq \sum_{u=0}^t \mu^{t-u} F\left(g_u^{(1)}, \ldots, g_u^{(n)}\right) \tag{1}$$

---

[1]Namely: $736 \times 5 = 3680$ seeded runs, and one single script to reproduce all of our results.

Function $F$ is called a *Gradient Aggregation Rule* (GAR). In non-Byzantine settings, averaging is used; formally: $F\left(g_t^{(1)}, \ldots, g_t^{(n)}\right) = \frac{1}{n} \sum_{i=1}^{n} g_t^{(i)}$. In the presence of Byzantine workers, a more robust aggregation is performed with a *Byzantine-resilient* GAR. Sections 2.2 and 2.3 respectively describe the 6 existing GARs and 2 attacks studied in this paper.

**Adversarial Model.** The goal of the adversary is to impede the learning process, which is defined as the maximization of the loss $Q$ or, more judiciously for the image *classification tasks* tackled in this paper, as the minimization[2] of the model's top-1 cross-accuracy. The adversary cannot directly overwrite $\theta_t$ at the parameter server. The adversary only submits $f$ arbitrary gradients to the server per step, via the $f$ Byzantine workers it controls[3]. We assume an omniscient adversary. In particular, the adversary knows the GAR used by the parameter server and, at each step, the adversary can generate Byzantine gradients dependent on the honest gradients submitted at the same step and any previous step.

## 2.2 Byzantine-resilient GARs

We briefly present below the 6 studied *Gradient Aggregation Rules* (GARs). These GARs are *Byzantine-resilient* (Section A), a notion first introduced by Blanchard et al. (2017) under the name $(\alpha, f)$-Byzantine-resilience. When used within its operating assumptions, a Byzantine-resilient GAR guarantees *convergence* even in an adversarial setting.

Let $n$ be the number of gradients the parameter server received from the $n$ workers (Figure 2), and let $f$ be the maximum number of Byzantine gradients the GAR must tolerate.

**Krum** (Blanchard et al., 2017). Each received gradient is assigned a score. The score of gradient $x$ is the sum of the squared $\ell_2$-distances between $x$ and the $n-f-2$ closest gradients to $x$. The aggregated gradient is then the arithmetic mean of the $n-f-2$ gradients with the smallest scores. This variant is called *Multi-Krum* in the original paper.

To be proven $(\alpha, f)$-Byzantine resilient, *Krum* requires the variance of the honest gradients $\mathbb{E} \left\| \mathcal{G}_t - \mathbb{E}\,\mathcal{G}_t \right\|^2$ to be bounded above as follows:

$$2 \cdot \left(n - f + \frac{f\,(n-f-2) + f^2\,(n-f-1)}{n - 2f - 2}\right) \cdot \mathbb{E} \left\| \mathcal{G}_t - \mathbb{E}\,\mathcal{G}_t \right\|^2 < \left\| \mathbb{E}\,\mathcal{G}_t \right\|^2 \tag{2}$$

**Median** (Yin et al., 2018). The coordinate-wise median of the $n$ received gradients. *Median* is proven $(\alpha, f)$-Byzantine resilience with the following condition on the *variance-norm* ratio:

$$(n - f) \cdot \mathbb{E} \left\| \mathcal{G}_t - \mathbb{E}\,\mathcal{G}_t \right\|^2 < \left\| \mathbb{E}\,\mathcal{G}_t \right\|^2 \tag{3}$$

**Trimmed Mean** (Yin et al., 2018). The coordinate-wise trimmed-mean of the $n$ received gradients. The *trimmed-mean* of $n$ values is the arithmetic mean, after the $f$ smallest and the $f$ largest values have been discarded, of the remaining values. From Theorem 1 of Xie et al. (2018b), we can derive the following condition on the *variance-norm* ratio:

$$\frac{2\,(f+1)\,(n-f)}{(n-2f)^2} \mathbb{E} \left\| \mathcal{G}_t - \mathbb{E}\,\mathcal{G}_t \right\|^2 < \left\| \mathbb{E}\,\mathcal{G}_t \right\|^2 \tag{4}$$

**Phocas** (Xie et al., 2018b). The coordinate-wise arithmetic mean of the $n-f$ closest values to the coordinate-wise trimmed-mean. From Theorem 2 of Xie et al. (2018b):

$$\left(4 + \frac{12\,(f+1)\,(n-f)}{(n-2f)^2}\right) \mathbb{E} \left\| \mathcal{G}_t - \mathbb{E}\,\mathcal{G}_t \right\|^2 < \left\| \mathbb{E}\,\mathcal{G}_t \right\|^2 \tag{5}$$

---

[2]Exempli gratia, with 10 classes, the worst possible final accuracy is arguably 0.1.

[3]Said otherwise, the $f$ Byzantine workers can collude.

**MeaMed** (Xie et al., 2018a). Same as *Phocas*, but with median replacing trimmed-mean. Theorem 5 of Xie et al. (2018a) provides the following condition:

$$10\,(n-f)\cdot\mathbb{E}\left\|\mathcal{G}_t - \mathbb{E}\,\mathcal{G}_t\right\|^2 < \left\|\mathbb{E}\,\mathcal{G}_t\right\|^2 \tag{6}$$

**Bulyan** (El-Mhamdi et al., 2018). This is a *composite* GAR, iterating on another GAR in a first selection phase. In the remaining of this paper, *Bulyan* will use *Krum*, so the first phase selects $n - 2\,f - 2$ gradients, at each iteration removing the highest scoring gradient. The aggregated gradient is the coordinate-wise arithmetic mean of the $n - 4\,f - 2$ closest values to the (coordinate-wise) median of the selected gradients.

The theoretical requirement on the *variance-norm* ratio are the same as the ones of the underlying GAR. That is, in this paper, they are the same as *Krum* (Equation 2).

### 2.3 Studied Attacks

The two state-of-the-art attacks, that recently appeared in the literature, follow the same core principle. Let $\varepsilon \in \mathbb{R}_{\geq 0}$ be a non-negative factor, and $a_t \in \mathbb{R}^d$ an *attack vector* which value depends on the actual attack used (see below for possible values of $a_t$). At each step $t$, each of the $f$ Byzantine workers submits the same Byzantine gradient: $\overline{g_t} + \varepsilon\,a_t$, where $\overline{g_t}$ is an approximation of the real gradient $\nabla Q\,(\theta_t)$ at step $t$. The value of $\varepsilon$ is fixed (see below).

**A Little is Enough** (Baruch et al., 2019). In this attack, each Byzantine worker submits $\overline{g_t} + \varepsilon\,a_t$, with $a_t \triangleq -\sigma_t$ the opposite of the coordinate-wise standard deviation of the honest gradient distribution $\mathcal{G}_t$. Our experiments use $\varepsilon = 1.5$, as proposed by the original paper.

**Fall of Empires** (Xie et al., 2019a). Each Byzantine worker submits $(1 - \varepsilon)\,\overline{g_t}$, i.e., $a_t \triangleq -\overline{g_t}$. The original paper tested $\epsilon \in \{-10, -1, 0, 0.1, 0.2, 0.5, 1, 10, 100\}$, our experiments use[4] $\varepsilon = 1.1$, corresponding in the notation of the original paper to $\epsilon \triangleq -(1 - \varepsilon) = -(1 - 1.1) = 0.1$.

## 3 Momentum at the Workers

Intuitively, the Byzantine-resilient GARs (Section 2.2) rely on the honest gradients being sufficiently *clumped* (formalized in e.g. Equation 2 to Equation 6). In the edge case where every honest gradient is equal (i.e. no stochastic noise), no attack can affect the learning: there is by assumption a strict majority of identical honest gradients. On the contrary when the honest gradients are "spread", i.e. their variance is *large enough* compared to their norms, the attack vectors can form a majority by relying on a few outlier (but honest) gradients (Baruch et al., 2019), and so substantially influence the aggregated gradient.

Momentum makes the parameters $\theta_t$ travel down the loss function with *inertia*, accumulating both the *real* gradient $\nabla Q\,(t)$ and the *error* (i.e. here, the *stochastic noise*) $g_t - \nabla Q\,(t)$. Intuitively, the accumulation of *errors* grows at a moderate rate, as past errors can be partially compensated by future ones. But when consecutive $\nabla Q\,(t)$ have sufficiently low solid angles, past real gradients do not compensate future real gradients: the *norm* of $G_t$ can grow "faster" (for each new step) than its *variance*, mitigating the potential impact of an attack.

### 3.1 Formulation

From the formulation of *momentum SGD* in a distributed setting (Equation 1):

$$G_t \triangleq \sum_{u=0}^{t} \mu^{t-u} F\left(g_u^{(1)}, \ldots, g_u^{(n)}\right)$$

we instead confer the momentum computation on the workers:

$$G_t \triangleq F\Bigg(\underbrace{\sum_{u=0}^{t} \mu^{t-u} g_u^{(1)}}_{G_t^{(1)}}, \ldots, \underbrace{\sum_{u=0}^{t} \mu^{t-u} g_u^{(n)}}_{G_t^{(n)}}\Bigg) \tag{7}$$

---

[4]This factor made this attack consistently successful in the original paper.

**Notations.** In the remaining of this paper, we call the original formulation *(momentum) at the server*, and the proposed, revised formulation *(momentum) at the worker(s)*. The quantities $G_t^{(1)} \ldots G_t^{(n)}$ will be called the *submitted gradients* (at step $t$). At step $t$, the *variance-norm* ratio is computed on the honest subset of: $g_t^{(1)} \ldots g_t^{(n)}$, if momentum at the server is employed, otherwise $G_t^{(1)} \ldots G_t^{(n)}$, if momentum at the workers is used instead.

**Formal analysis.** The formal analysis of the impact of our technique on the *variance-norm* ratio of the aggregated gradients is available in the appendix, Section B.

## 4 EXPERIMENTS

Our experiments cover 2 models, 4 datasets, the 6 studied defenses under each of the 2 state-of-the-art attacks[5], different fractions of Byzantine workers (either *half* or *a quarter*), using Nestorov instead of classical momentum, plus unattacked settings where each worker is honest and the GAR is mere *averaging*. Since our theoretical results (Section B) suggest that smaller learning rates may reduce the *variance-norm* ratio, two learning rate schedules (an *optimal* and a *smaller* one) are also tested. For reproducibility and confidence in the empirical benefits of our reformulation, we test every combination of the hyperparameters mentioned above, and each combination is repeated 5 times with specified *seeds* (1 to 5, totally 3680 runs).

The tools we developed to implement our reformulation captures $\sim 20$ metrics, including the evolution of the average loss, top-1 cross-accuracy and *variance-norm* ratio of the submitted gradients. In this section and Section E, we specifically report on these 3 metrics.

### 4.1 EXPERIMENTAL SETUP

We use a compact notation to define the models: L(#outputs) for a fully-connected linear layer, R for ReLU activation, S for log-softmax, C(#channels) for a fully-connected 2D-convolutional layer (kernel size 3, padding 1, stride 1), M for 2D-maxpool (kernel size 2), B for batch-normalization, and D for dropout (with fixed probability 0.25).

We use the models from respectively Baruch et al. (2019) and Xie et al. (2019a):

|  | *Fully connected* | *Convolutional* |
|---|---|---|
| **Model** | (784)-L(100)-R-L(10)-R-S | (3, 32×32)-C(64)-R-B-C(64)-R-B-M-D- -C(128)-R-B-C(128)-R-B-M-D- -L(128)-R-D-L(10)-S |
| **Datasets** | MNIST, Fashion MNIST (83 samples/gradient) | CIFAR-10, CIFAR-100 (50 samples/gradient) |
| **#workers** | $n = 51$   $f \in \{24, 12\}$ | $n = 25$   $f \in \{11, 5\}$ |

For model training, we use the *negative log likelihood* loss and respectively $10^{-4}$ and $10^{-2}$ $\ell_2$-regularization for the *fully connected* and *convolutional* models. We also clip gradients, ensuring their norms remain respectively below 2 and 5 for the *fully connected* and *convolutional* models. Regarding evaluation, we use the *top-1 cross-accuracy* over the whole test set.

Datasets are pre-processed before training. MNIST receives the same pre-processing as in Baruch et al. (2019): an input image normalization with mean 0.1307 and standard deviation 0.3081. Fashion MNIST, CIFAR-10 and CIFAR-100 are all expanded with horizontally flipped images. For both CIFAR-10 and CIFAR-100, a per-channel normalization with means 0.4914, 0.4822, 0.4465 and standard deviations 0.2023, 0.1994, 0.2010 (Liu, 2019) has been applied.

We denote by $f$ the number of Byzantine workers either to the maximum for which *Krum* can be used (roughly an half: $f = \lfloor \frac{n-3}{2} \rfloor$), or the maximum for *Bulyan* (roughly a quarter, $f = \lfloor \frac{n-3}{4} \rfloor$). The attack factors $\varepsilon_t$ (Section 2.3) are set to constants proposed in the literature, namely $\varepsilon_t = 1.5$ for Baruch et al. (2019) and $\varepsilon_t = 1.1$ for Xie et al. (2019a).

---

[5]To the best of our knowledge, putting aside simple attacks (e.g. sending attack gradients sampled from a Gaussian distribution) tested in each defense papers, no other attack has been published.

We also experiment two different learning rates. The first and largest is selected so as to maximize the *performance* (highest final cross-accuracy and accuracy gain per step) of the model trained without Byzantine workers. The second and smallest is chosen so as to minimize the *performance loss* under attack, without substantially impacting the final accuracy when trained without Byzantine workers. The *fully connected* and *convolutional* models are trained respectively with $\mu = 0.9$ and $\mu = 0.99$. These values were obtained by trial and error, to maximize the accuracy gain per step when there is no attack.

**Reproducibility.** Particular care has been taken to make our results reproducible. Each of the 5 runs per experiment are respectively seeded with seed 1 to 5. For instance, this implies that two experiments with same seed and same model also starts with the same parameters $\theta_0$. To further reduce the sources of non-determinism, the CuDNN backend is configured in deterministic mode (our experiments ran on two *GeForce GTX 1080 Ti*) with benchmark mode turned off. We also used *log-softmax + nll loss*, which is equal to *softmax + cross-entropy loss*, but with improved numerical stability on PyTorch. We provide our code along with a script reproducing *all* of our results, both the experiments and the graphs, in one command. Details, including software and hardware dependencies, are available in Section C.

## 4.2 Experimental Results

This section reports on the evolution of the average loss, top-1 cross-accuracy and *variance-norm* ratio of the submitted gradients. Section E in the appendix reports on the entirety of our experimental results, and Section D additionally experiments with a much larger model.

One first important remark is that our new formulation either obtain similar, or (subtantially) increased maximum top-1 cross-accuracy measured, compared to the standard formulation in the exact same settings. Namely, in only 4 pairs of runs (0.23% of all the tested pairs) did our formulation lead to a decreased maximum top-1 cross-accuracy. Also, these decreases were only observed with the *fully connected* model, using *Krum* against Xie et al. (2019a), and for each of these 4 runs using any of the 4 other seeds made the decrease disappear.

In all of our experiments, we observe a strong correlation between higher top-1 cross-accuracies and lower average losses; e.g. see Figure 4. The two state-of-the-art attacks decreased the accuracy by at least 20%, compared to the unattacked case (see "No attack" in Figure 3), in 25.80% and 70.80% of the runs with respectively the *fully connected* and *convolutional* models.

Focusing on the *convolutional* model, when roughly *an half* of the workers are Byzantine, both attacks actually succeed in decreasing the accuracy by at least 20% in 100% of our runs. Our technique manages to recover at least 10% and 20% in respectively 79.75% and 49.25% of these runs. When roughly *a quarter* of the workers are Byzantine, the attacks decrease the accuracy by at least 20% in 46.46% of our runs. Our technique then manages to recover at least 20% in 95.07% of these runs. Figure 3 shows a fraction of these runs.

Technically, our reformulation aims at reducing the *variance-norm* ratio of the aggregated gradients. Intuitively, this ratio is expected to increase as the loss decreases; more correctly as the norm of the gradient decreases. For instance, Figure 5 displays the *variance-norm* ratios of *Trimmed Mean* and *Bulyan* using the same settings as in Figure 4. At least before the final cross-accuracy is reached, our technique consistently decreases the *variance-norm* ratio of the aggregated gradients. Also, we consistently observed in the experiments that reducing the learning rate indeed reduces the *variance-norm* ratio (e.g. Figure 5, $t \geq 1500$).

## 5 Related and future work

**Alternative Byzantine-resilient Approaches.** The Byzantine abstraction is a very general fault model that has long been studied in distributed computing (Lamport et al., 1982). The standard, golden solution for Byzantine fault tolerance is the *state machine replication* approach (Schneider, 1990). This approach is however based on *replication*, which is known to be unsuitable for distributed machine learning and stochastic gradient descent.

Chen et al. (2018) was the first Byzantine-resilient mechanism based on a *redundancy* scheme rather than statistical robustness. While the proposed mechanism is not vulnerable to the

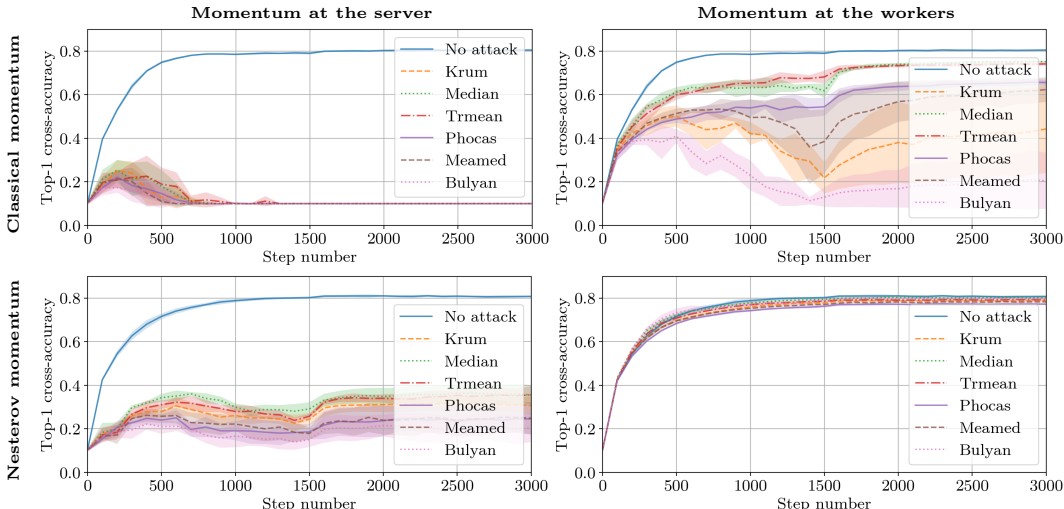

Figure 3: CIFAR-10 and the *convolutional* model (Section 4.1), with $n = 25$, $f = 5$ and $\alpha_t = 0.01$ if $t < 1500$ else $\alpha_t = 0.001$, under attack from (Baruch et al., 2019). Each line and colored surface correspond to respectively the average and standard deviation of the top-1 cross-accuracy over 5 seeded runs. Only two parameters change between graphs: where momentum is computed (*at the server* or *at the workers*), and which flavor of momentum is employed.

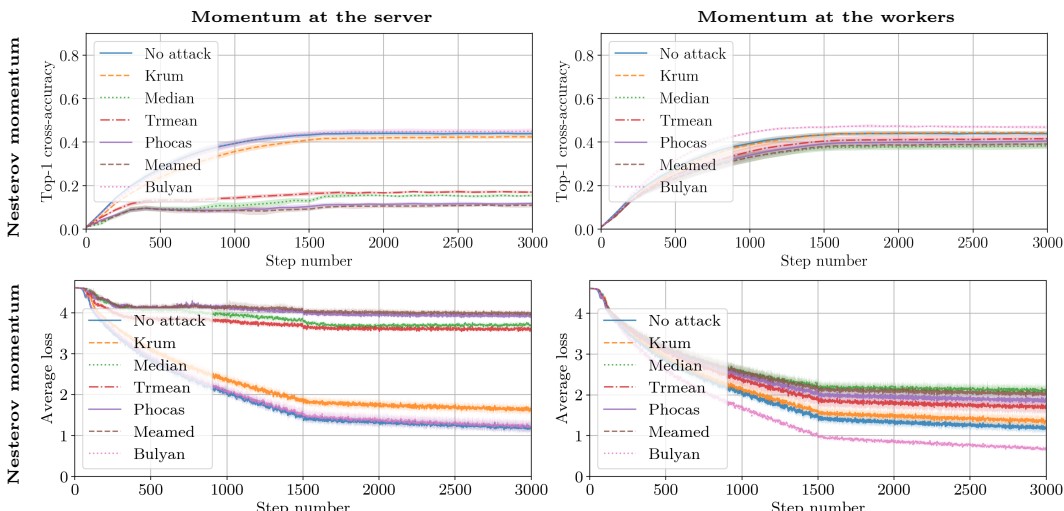

Figure 4: Accuracy and average loss, CIFAR-100 and the *convolutional* model, with $n = 25$, $f = 5$ and $\alpha_t = 0.01$ if $t < 1500$ else $\alpha_t = 0.001$, under attack from (Xie et al., 2019a).

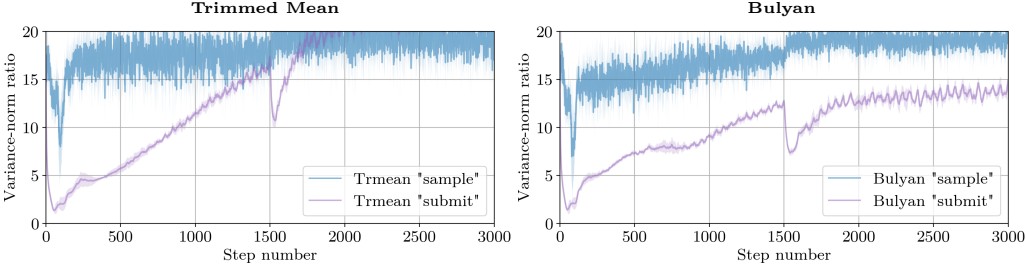

Figure 5: Same settings as in Figure 4, *variance-norm* ratios of *Trimmed Mean* and *Bulyan* with *momentum at the workers*. "sample" corresponds to the *variance-norm* ratio of the sampled gradients, and "submit" to the *variance-norm* ratio of the *submitted gradients*.

attacks discussed in this paper, it induces (due to the redundancy) substantial computational costs compared to statistically-robust techniques. Rajput et al. (2019) combines Chen et al. (2018) with statistically-robust GARs into a *hybrid* system, and achieves an improved aggregation time. Under the requirements of both redundancy-based schemes and statistically-robust GARs, Rajput et al. (2019) can significantly decrease the voting power of the adversary, and can consequently also deflect attacks in these cases. Xie et al. (2019b), and its follow-up (Xie, 2019), introduced the concept of *suspicion-based* fault-tolerance: each gradient is assigned a score based on the loss obtained by descending with this gradient only. The lowest scoring gradients are then filtered-out, and the remaining gradients are averaged and used to update the model. Compared to *statistically-robust* approaches, such a scheme does not rely on sufficiently low variance-norm ratios, and gradients could be processed independently. This advantage comes at the expense of a substantial computational load on the parameter server, as the server has to compute several forward passes for each received gradients, basically doing half the work of each worker again (the workers also have backpropagation passes). Most importantly, the success of such a technique is conditioned to the quality of the loss estimations at the server: if the estimation is biased (e.g. the server's dataset is "different" than the workers' ones) , has a high variance (e.g. the server's batch size is "too" small), or if the "hard threshold" $\epsilon$ used is too large (would accept Byzantine gradients)/too small (would refuse even honest gradients) the defense might be ineffective/harmful.

**Momentum-based Variance Reduction.** Our algorithm is different from Cutkosky & Orabona (2019), as instead of reducing the variance of the gradients, we actually *increase* it (Equation 8). What we seek to reduce is the *variance-norm* ratio, which is the key quantity for any Byzantine-resilient GAR approximating a high-dimensional median, e.g. *Krum*, *Median*, as well as Yang & Bajwa (2019b;a); Chen et al. (2017); Muñoz-González et al. (2019)[6]. Some of the ideas introduced in Cutkosky & Orabona (2019) could nevertheless help further improve Byzantine resilience. For instance, introducing an adaptive learning rate which decreases depending on the curvature of the parameter trajectory is an appealing approach to further reduce the *variance-norm* ratio (Equation 10). The computation of momentum *at the workers* has also been used in the literature for the purpose of *gradient compression* (Lin et al., 2018). These techniques are nevertheless not (meant to be) Byzantine-resilient.

**Future Work.** The theoretical condition to reduce the *variance-norm* ratio of the submitted gradients (compared to the *variance-norm* ratio of the sampled gradients at the same step), in Section B, shows that momentum at the workers is a double-edged sword. The problem is that $s_t$ can become negative: the norm of the momentum gradient would then be *decreased*, increasing the *variance-norm* ratio. While the ability to cross narrow, local minima is recognized as an accelerator (Goh, 2017), for the purpose of Byzantine-resilience we want to ensure momentum at the workers does not increase the *variance-norm* ratio (compared to the *variance-norm* ratio of the sampled gradients at the same step). The theoretical condition for this purpose is given in Equation 9. One simple amendment would then be to use momentum at the workers when Equation 9 is satisfied, and fallback to computing it at the server otherwise. Also, a more complex, possible future approach could be to dynamically adapt the momentum factor $\mu$, decreasing it as the curvature increases.

**Asynchronous SGD.** We focused in this work on the synchronous setting, which received most of the attention in the Byzantine-resilient literature. Yet, we do not see any issue that would prevent our work from being applied in asynchronous settings. Specifically, combining our idea with a filtering scheme such as *Kardam* (Damaskinos et al., 2018) is in principle possible, as this filter and momentum commute. However, further analysis of the interplay between the dynamics of stale gradients and the dynamics of momentum remain necessary.

**Byzantine Servers.** While most of the research on Byzantine-resilience gradient descent has focused on the workers' side, assuming a reliable server, recent efforts have started tackling Byzantine servers (El-Mhamdi et al., 2020). Our reduction of the *variance-norm* ratio strengthens the gradient aggregation phase, which is necessary whether we deal with Byzantine workers or Byzantine servers. An interesting open question is whether the dynamics of momentum could positively affect the model drift between different parameter servers in a Byzantine context. Any quantitative answer to this question could enable the use of our method in fully decentralised Byzantine resilient gradient descent.

## 6 Acknowledgments

This work has been supported in part by the Swiss National Science Foundation (FNS grant N°200021_182542).

We would also like to acknowledge here and thank the anonymous individuals who partook in the review of this work and its code, in this final version and previous drafts, for their valuable time and inputs.

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

## A  Byzantine Resilience: Definition

Let $n$ be the number of gradients the parameter server received from the $n$ workers (Figure 2), and let $f$ be the maximum number of Byzantine gradients the GAR must tolerate. Some other notations below are reused from Section 2.1.

**Definition 1.** Without loss of generality, let $g_t^{(1)} \ldots g_t^{(n-f)} \sim \mathcal{G}_t$ be $n - f$ independent, "honest" gradients following the same distribution $\mathcal{G}_t$, and let $g_t^{(n-f+1)} \ldots g_t^{(n)} \in \left(\mathbb{R}^d\right)^f$ be arbitrary gradients, each possibly dependent on $\mathcal{G}_t$ and the "honest" gradients. A GAR $F$ is said $(\alpha, f)$-Byzantine resilient if and only if $g_t \triangleq F\left(g_t^{(1)}, \ldots, g_t^{(n)}\right)$ satisfies:

1. $\left\langle \mathbb{E}\, g_t, \mathbb{E}\, g_t^{(1)} \right\rangle > 0$
2. $\forall r \in \{2, 3, 4\}$, $\mathbb{E} \|g_t\|^r$ is bounded above by a linear combination of the terms $\mathbb{E} \left\| g_t^{(1)} \right\|^{r_1} \ldots \mathbb{E} \left\| g_t^{(1)} \right\|^{r_k}$, with $(k, r_1 \ldots r_k) \in (\mathbb{N}^*)^{k+1}$ and $r_1 + \ldots + r_k = r$.

## B  Momentum at the Workers: Effects

We compare the *variance-norm* ratio of the non-Byzantine subset of the sampled gradients $g_t^{(1)} \ldots g_t^{(n)}$ against the *variance-norm* ratio of the non-Byzantine subset of the submitted gradients $G_t^{(1)} \ldots G_t^{(n)}$ when *classical* momentum is computed *at the workers*.

We denote by $\mathbb{E}\, \mathcal{G}_t \triangleq \nabla Q\left(\theta_t\right)$ the "real" gradient[7] at step $t$.

Let $\lambda_t \triangleq \|\mathbb{E}\, \mathcal{G}_t\| > 0$ be the real gradient's norm at step $t$.

Let $\sigma_t \triangleq \sqrt{\mathbb{E} \|\mathcal{G}_t - \mathbb{E}\, \mathcal{G}_t\|^2}$ be the standard deviation of the real gradient at step $t$. The *variance-norm* ratio of the non-Byzantine subset of the sampled gradients at step $t$ is:

$$r_t^{(s)} \triangleq \frac{\sigma_t^2}{\lambda_t^2}$$

---

[7]In this analysis, the expectations are by default conditioned on the past randomness (all what happened up to step $t$) from the $(n - f) \cdot b$ data-points sampled, at each past step, by the $(n - f)$ honest workers to estimate their respective gradients.

We will now compute the *variance-norm* ratio of the non-Byzantine subset of the submitted gradients. Let $G_t^{(i)}$, with $G_{-1}^{(i)} \triangleq 0$, be the gradient sent by any honest worker $i$ at step $t$, i.e.:

$$G_t^{(i)} \triangleq \sum_{u=0}^{t} \mu^{t-u} g_u^{(i)}$$

The numerator of the *variance-norm* ratio is, for any two honest worker identifiers $i \neq j$:

$$\mathbb{E} \left\| G_t^{(i)} - G_t^{(j)} \right\|^2$$

$$= \mathbb{E} \left\| g_t^{(i)} + \mu\, G_{t-1}^{(i)} - g_t^{(j)} - \mu\, G_{t-1}^{(j)} \right\|^2$$

$$= \mathbb{E} \left\| g_t^{(i)} - g_t^{(j)} \right\|^2 + \mu^2\, \mathbb{E} \left\| G_{t-1}^{(i)} - G_{t-1}^{(j)} \right\|^2 + 2\,\mu\, \underbrace{\left( \underbrace{\mathbb{E}\, g_t^{(i)} - \mathbb{E}\, g_t^{(j)}}_{=\,\mathbb{E}\, \mathcal{G}_t - \mathbb{E}\, \mathcal{G}_t} \right) \cdot \left( \mathbb{E}\, G_{t-1}^{(i)} - \mathbb{E}\, G_{t-1}^{(j)} \right)}_{=\,0}$$

$$= \mathbb{E} \left\| g_t^{(i)} - g_t^{(j)} \right\|^2 + \mu^2\, \mathbb{E} \left\| G_{t-1}^{(i)} - G_{t-1}^{(j)} \right\|^2$$

$$= 2\,{\sigma_t}^2 + \mu^2 \left( 2\,{\sigma_{t-1}}^2 + \mu^2 \left( 2\,{\sigma_{t-2}}^2 + \mu^2\, (...) \right) \right)$$

$$= 2 \sum_{u=0}^{t} \mu^{2(t-u)} {\sigma_u}^2 \tag{8}$$

$$= 2\, \mathbb{E} \left\| G_t^{(i)} - \mathbb{E}\, G_t^{(i)} \right\|^2$$

And the denominator of the *variance-norm* ratio is:

$$\left\| \mathbb{E}\, G_t^{(i)} \right\|^2 = \left\| \mathbb{E}\, g_t^{(i)} + \mu\, \mathbb{E}\, G_{t-1}^{(i)} \right\|^2$$

$$= \left\| \mathbb{E}\, g_t^{(i)} \right\|^2 + 2\,\mu\, \mathbb{E}\, g_t^{(i)} \cdot \mathbb{E}\, G_{t-1}^{(i)} + \mu^2 \left\| \mathbb{E}\, G_{t-1}^{(i)} \right\|^2$$

$$= {\lambda_t}^2 + 2\,\mu\, \mathbb{E}\, g_t^{(i)} \cdot \left( \mathbb{E}\, g_{t-1}^{(i)} + \mu \left( \mathbb{E}\, g_{t-2}^{(i)} + \mu\, (...) \right) \right)$$

$$+ \mu^2 \left( {\lambda_{t-1}}^2 + 2\,\mu\, \mathbb{E}\, g_{t-1}^{(i)} \cdot \left( \mathbb{E}\, g_{t-2}^{(i)} + \mu\, (...) \right) \right.$$

$$\left. + \mu^2\, \mathbb{E} \left\| G_{t-2}^{(i)} \right\|^2 \right)$$

$$= \sum_{u=0}^{t} \mu^{2(t-u)} \left( {\lambda_u}^2 + 2 \sum_{v=0}^{u-1} \mu^{u-v} \underbrace{\mathbb{E}\, g_u^{(i)} \cdot \mathbb{E}\, g_v^{(i)}}_{=\,\mathbb{E}\, \mathcal{G}_u \cdot \mathbb{E}\, \mathcal{G}_v} \right)$$

Thus, assuming honest gradients $\mathbb{E}\, G_t^{(i)}$ do not become null:

$$r_t^{(w)} \triangleq \frac{{\Omega_t}^2}{{\Lambda_t}^2} = \frac{\sum_{u=0}^{t} \mu^{2(t-u)} {\sigma_u}^2}{\sum_{u=0}^{t} \mu^{2(t-u)} \left( {\lambda_u}^2 + s_u \right)}$$

where the expected "straightness" of the gradient trajectory at step $u$ is defined by:

$$s_u \triangleq 2 \sum_{v=0}^{u-1} \mu^{u-v} \mathbb{E}\, \mathcal{G}_u \cdot \mathbb{E}\, \mathcal{G}_v$$

$s_u$ quantifies what is intuitively the *curvature* of the gradient trajectory. Straight trajectories can make $s_u$ grow up to $(1-\mu)^{-1} > 1$ times the expected squared-norm of the honest gradients, while highly "curved" trajectories (e.g. close to a local minimum) can make $s_u$ negative.

This observation stresses that this formulation of momentum can sometimes be harmful for the purpose of Byzantine resilience. We measured $s_u$ for every step $u > 0$ in our experiments, and we always observed that this quantity is positive and increases for a short window of (dozen) steps (depending on $\alpha_t$), and then oscillates between positive and negative values. While the empirical impact (decreased or cancelled loss in accuracy) is concrete, we believe there is room for further improvements, as discussed in Section 5.

The purpose of using momentum at the workers is to reduce the *variance-norm* ratio $r_t^{(w)}$, compared to $r_t^{(s)}$. Since $g_0^{(i)} = G_0^{(i)}$, we verify that $r_0^{(u)} = r_0^{(w)}$. Then $\forall t > 0$, assuming $\Omega_{t-1} > 0$ and $\sigma_t > 0$, we have:

$$
\begin{aligned}
r_t^{(w)} \leq r_t^{(s)} &\Leftrightarrow \frac{\sigma_t^2 + \mu^2 \Omega_{t-1}^2}{\lambda_t^2 + s_t + \mu^2 \Lambda_{t-1}^2} \leq \frac{\sigma_t^2}{\lambda_t^2} \\
&\Leftrightarrow \mu^2 \Omega_{t-1}^2 \lambda_t^2 \leq \left( s_t + \mu^2 \Lambda_{t-1}^2 \right) \sigma_t^2 \\
&\Leftrightarrow s_t \geq \mu^2 \Lambda_{t-1}^2 \left( \frac{r_{t-1}^{(w)}}{r_t^{(s)}} - 1 \right)
\end{aligned}
\tag{9}
$$

The condition for decreasing $r_t^{(w)}$ can be obtained similarly, assuming $\Omega_{t-1} > 0$ and $\sigma_t > 0$:

$$
r_t^{(w)} \leq r_{t-1}^{(w)} \Leftrightarrow s_t \geq \lambda_t^2 \left( \frac{r_t^{(s)}}{r_{t-1}^{(w)}} - 1 \right)
$$

To study the impact of a lower learning rate $\alpha_t$ on $s_t$, we will assume that the real gradient $\nabla Q$ is $l$-Lipschitz. Namely:

$$
\forall (t, u) \in \mathbb{N}^2, u < t, \| \mathbb{E} \, \mathcal{G}_t - \mathbb{E} \, \mathcal{G}_u \|^2 \leq l^2 \| \theta_t - \theta_u \|^2 \leq l^2 \left\| \sum_{v=u}^{t-1} \alpha_v \, G_v \right\|^2
$$

Then, $\forall (t, u) \in \mathbb{N}^2, u < t$, we can rewrite:

$$
\| \mathbb{E} \, \mathcal{G}_t - \mathbb{E} \, \mathcal{G}_u \|^2 = \underbrace{\| \mathbb{E} \, \mathcal{G}_t \|^2}_{\lambda_t^2} + \underbrace{\| \mathbb{E} \, \mathcal{G}_u \|^2}_{\lambda_u^2} - 2 \, \mathbb{E} \, \mathcal{G}_t \cdot \mathbb{E} \, \mathcal{G}_u
$$

And finally, we can lower-bound $s_t$ as:

$$
\begin{aligned}
&\sum_{u=0}^{t-1} \mu^{t-u} \| \mathbb{E} \, \mathcal{G}_t - \mathbb{E} \, \mathcal{G}_u \|^2 \\
&= \sum_{u=0}^{t-1} \mu^{t-u} \left( \lambda_t^2 + \lambda_u^2 \right) - 2 \underbrace{\sum_{u=0}^{t-1} \mu^{t-u} \, \mathbb{E} \, \mathcal{G}_t \cdot \mathbb{E} \, \mathcal{G}_u}_{s_t} \\
&\leq \sum_{u=0}^{t-1} \mu^{t-u} \, l^2 \left\| \sum_{v=u}^{t-1} \alpha_v \, G_v \right\|^2 \\
&\Leftrightarrow s_t \geq \sum_{u=0}^{t-1} \mu^{t-u} \left( \lambda_t^2 + \lambda_u^2 - l^2 \left\| \sum_{v=u}^{t-1} \alpha_v \, G_v \right\|^2 \right) \\
&\geq \frac{1 - \mu^t}{1 - \mu} \lambda_t^2 + \sum_{u=0}^{t-1} \mu^{t-u} \left( \lambda_u^2 - l^2 \left\| \sum_{v=u}^{t-1} \alpha_v \, G_v \right\|^2 \right)
\end{aligned}
\tag{10}
$$

When the real gradient $\nabla Q$ is (locally) Lipschitz continuous, reducing the learning rate $\alpha_t$ can suffice to ensure $s_t$ satisfies the conditions laid above for decreasing the *variance-norm* ratio $r_t^{(w)}$; the purpose of momentum at the workers. Importantly this last lower bound, namely Equation 10, sets how the practitioner should choose two hyperparameters, $\mu$ and $\alpha_t$, for the purpose of Byzantine-resilience. Basically, as long as it does not harm the training without adversary, $\mu$ should be set *as high* and $\alpha_t$ *as low* as possible.

## C    Reproducing the results

Our contributed code is available at `https://github.com/LPD-EPFL/ByzantineMomentum`, or as a ZIP archive from OpenReview (`https://openreview.net/forum?id=H8UHdhWG6A3`).

**Software dependencies.** Python 3.7.3 has been used, over several GNU/Linux distributions (Debian 10, Ubuntu 18). Besides the standard libraries associated with Python 3.7.3, our scripts also depend on[8]:

| Library | Version | Library | Version | Library | Version |
|---------|---------|----------|------------|-----------|--------|
| numpy | 1.19.1 | requests | 2.21.0 | pytz | 2020.1 |
| torch | 1.6.0 | urllib3 | 1.24.1 | dateutil | 2.8.1 |
| torchvision | 0.7.0 | chardet | 3.0.4 | pyparsing | 2.2.0 |
| pandas | 1.1.0 | certifi | 2018.08.24 | cycler | 0.10.0 |
| matplotlib | 3.0.2 | idna | 2.6 | kiwisolver | 1.0.1 |
| PIL | 7.2.0 | six | 1.15.0 | cffi | 1.13.2 |

**Hardware dependencies.** We list below the hardware components used:

- 1   Intel(R) Core(TM) i7-8700K CPU @ 3.70GHz
- 2   Nvidia GeForce GTX 1080 Ti
- 64 GB of RAM

### C.1    Command

Our results are reproducible in one command. In the root directory of the ZIP file:

```
$ python3 reproduce.py
```

On our hardware, reproducing the results takes a bit less than a week. Please be aware this script will require non-negligible disk space: 2.1 GB of run data, and 132 MB of graphs.

Depending on the hardware, instructing the script to launch several runs per available GPU may reduce the total runtime. For instance, to push up to 4 concurrent runs per GPU:

```
$ python3 reproduce.py --supercharge 4
```

## D    Larger models

To assess our method on even larger models, we consider the "wide-resnet" model family implemented by Kim (2020). We use the same model-specific parameters as the ones proposed by the original author, namely: 28 (depth), 10 (widen factor), 0.3 (dropout rate), and 10 output classes (for CIFAR-10). This model contains $36\,489\,290$ trainable parameters, almost 28 times more than the $1\,310\,922$ trainable parameters of the *convolutional* model.

We employ the same hyperparameters as in our main experiments with the *convolutional* model (Section 4.1), except for the number of workers (set to $n = 11$), the mini-batch size per worker (set to 20), and the learning rate schedule (0.02 for $t < 8000$, 0.004 for $8000 \leq t < 16000$, 0.0008 for $t \geq 16000$).

The training procedure lasts for $20\,000$ steps and only employs Nesterov momentum, as proposed by the original author (Kim, 2020). We report on the maximum observed top-1 cross-accuracy in Figure 6 and evolution of the top-1 cross-accuracy in figures 7 and 8.

These results are also reproducible in one command. In the root directory of the ZIP file:

```
$ python3 reproduce-appendix.py
```

On our hardware, reproducing these results takes several weeks. Some of the 6 presented GARs could not be employed, as they repeatedly trigger *out-of-memory* errors on our GPUs.

---

[8]This list was automatically generated (see `get_loaded_dependencies()` in `tools/misc.py`).

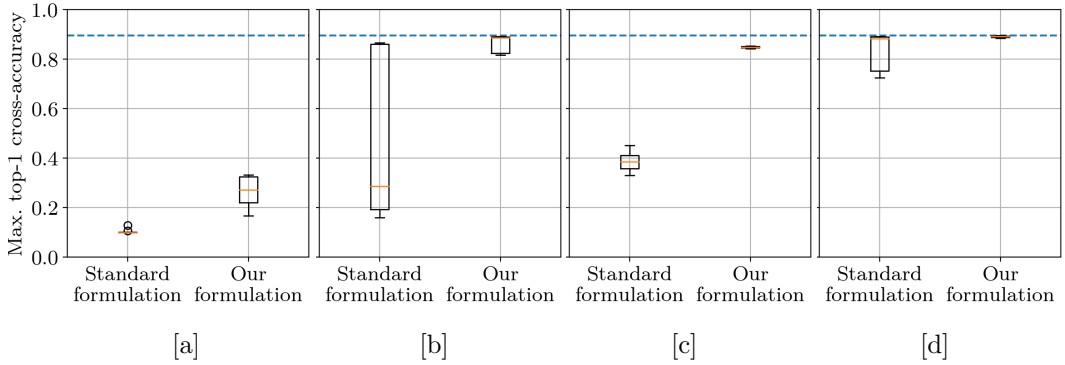

Figure 6: CIFAR-10 and *wide-resnet* model. [a] Roughly an half ($f = 4$) Byzantine workers implementing (Baruch et al., 2019). [b] Roughly a quarter ($f = 2$) Byzantine workers implementing (Baruch et al., 2019). [c] Roughly an half Byzantine workers implementing (Xie et al., 2019a). [d] Roughly a quarter Byzantine workers implementing (Xie et al., 2019a).

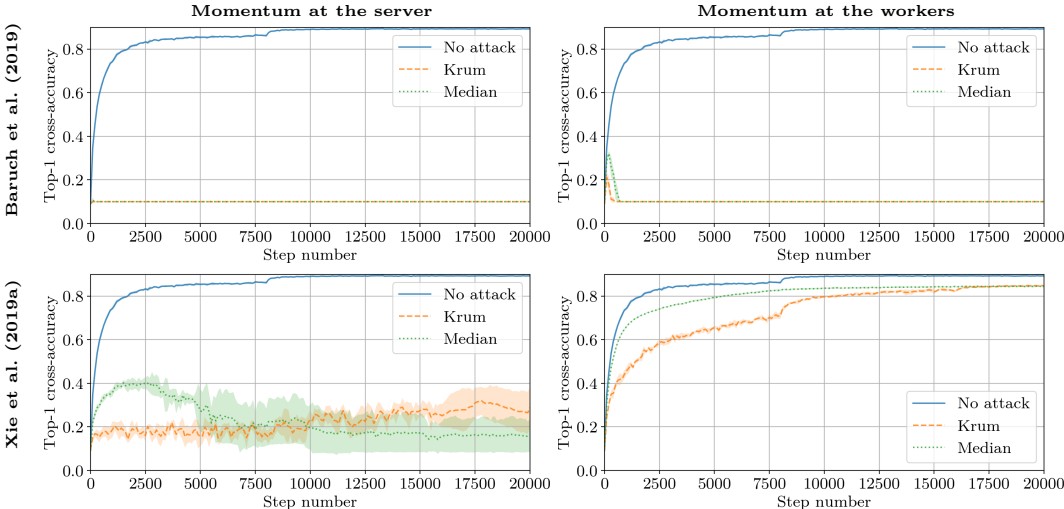

Figure 7: CIFAR-10 and *wide-resnet* model, roughly an half of Byzantine workers.

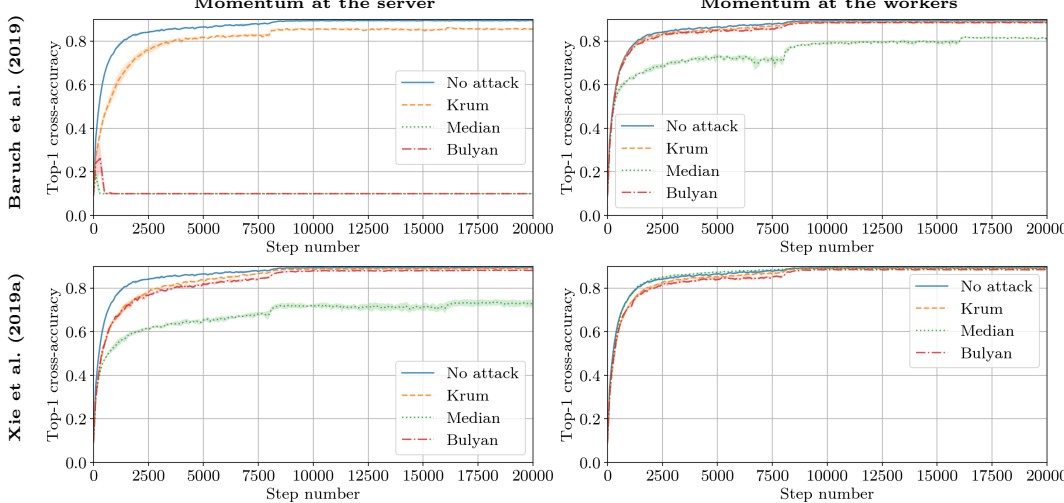

Figure 8: CIFAR-10 and *wide-resnet* model, roughly a quarter of Byzantine workers.

# E    More experimental results

This section reports on the entirety of the main experiments, completing Section 4 of the main paper. For every pair model-dataset, the following parameters vary:

- Which attack: Baruch et al. (2019) or Xie et al. (2019a)
- Which defense: *Krum*, *Median*, *Trimmed Mean*, *Phocas*, *MeaMed*, or *Bulyan*
- How many Byzantine workers (*an half* or *a quarter*)
- Where momentum is computed (*server* or *workers*)
- Which flavor of momentum is used (*classical* or *Nesterov*)
- Which learning rate is used (*larger* or *smaller*)

Every possible combination is tested[9], leading to a total of 736 different experiment setups. Each setup is tested 5 times, each run with a fixed seed from 1 to 5, enabling verbatim reproduction of our results[10]. In this specific section, we report on:

- the maximum observed top-1 cross-accuracy with each of the 6 studied GARs,
- the evolution of the average and standard deviation of the *top-1 cross-accuracy* for every tested setup.

The results regarding the maximum observed top-1 cross-accuracy are layed out by "block" of 4 experiment setups, among which only the flavor of momentum and the attack used are different. Namely: [a] classical momentum under attack from Baruch et al. (2019), [b] nesterov momentum under attack from Baruch et al. (2019), [c] classical momentum under attack from Xie et al. (2019a), [d] nesterov momentum under attack from Xie et al. (2019a).

The results regarding the evolution of the top-1 cross-accuracy are layed out by "blocks" of 4 experiment setups presenting the same model, dataset, learning rate schedule, number of Byzantine workers and attack. These results are presented from Figure 25 to Figure 56.

---

[9]Along with baselines using *averaging* without attack.
[10]Despite our best efforts, there may still exist minor sources of non-determinism, like race-conditions in the evaluation of certain functions (e.g., parallel additions) in a GPU. Nevertheless we believe these should not affect the results in any significant way.

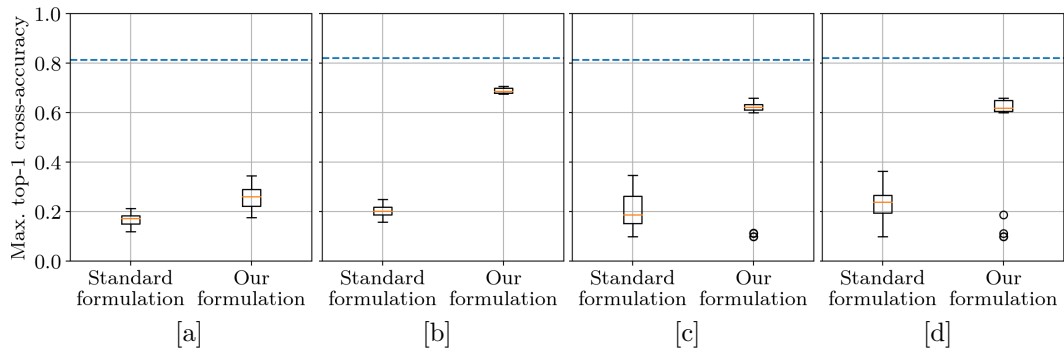

Figure 9: CIFAR-10 and *convolutional* model, with $n = 25$, $f = 11$ and $\alpha_t = 0.01$ if $t < 1500$ else $\alpha_t = 0.001$.

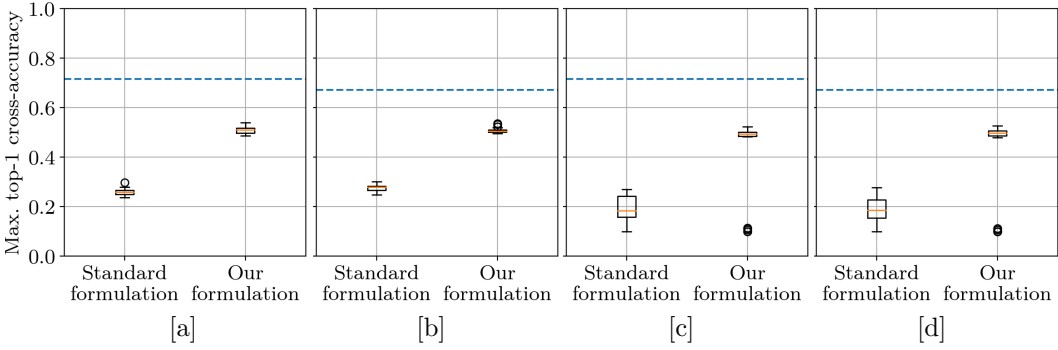

Figure 10: CIFAR-10 and *convolutional* model, with $n = 25$, $f = 11$ and $\alpha_t = 0.001$.

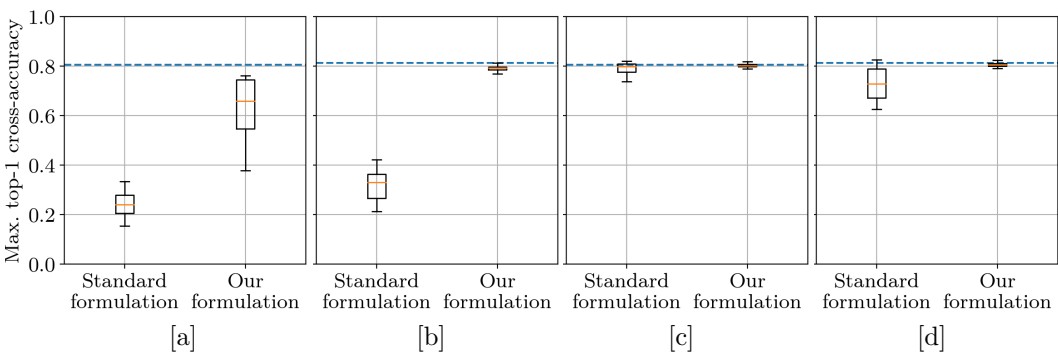

Figure 11: CIFAR-10 and *convolutional* model, with $n = 25$, $f = 5$ and $\alpha_t = 0.01$ if $t < 1500$ else $\alpha_t = 0.001$.

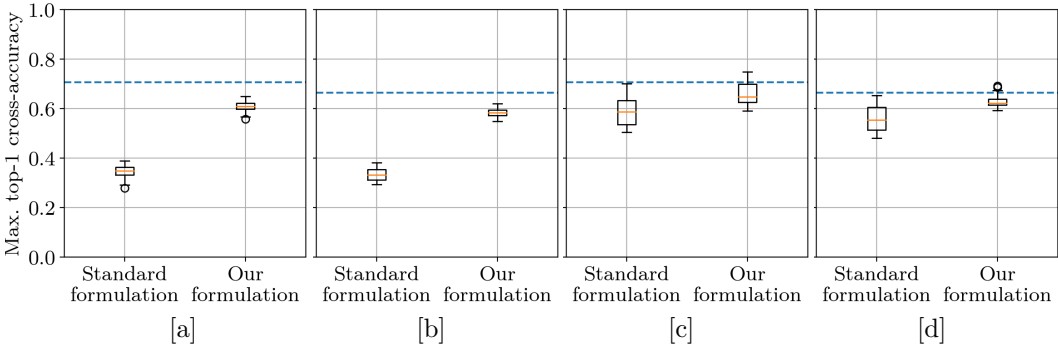

Figure 12: CIFAR-10 and *convolutional* model, with $n = 25$, $f = 5$ and $\alpha_t = 0.001$.

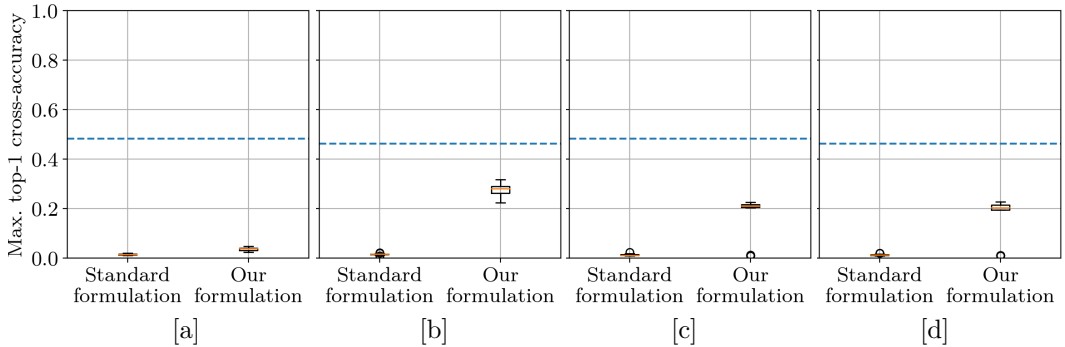

Figure 13: CIFAR-100 and *convolutional* model, with $n = 25$, $f = 11$ and $\alpha_t = 0.01$ if $t < 1500$ else $\alpha_t = 0.001$.

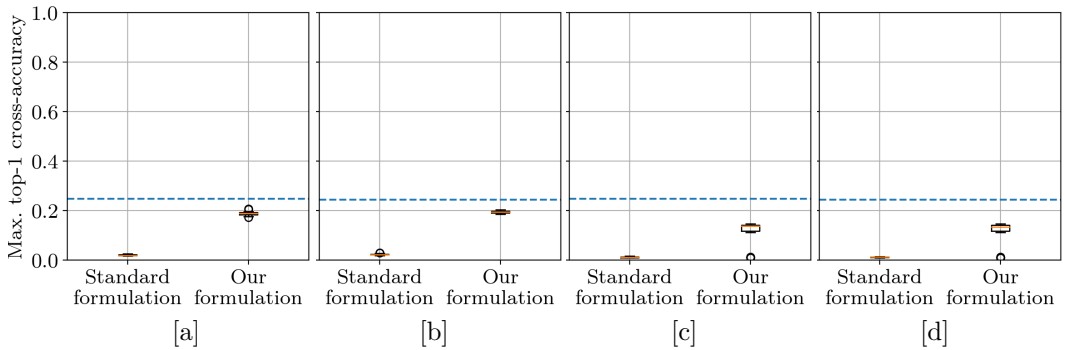

Figure 14: CIFAR-100 and *convolutional* model, with $n = 25$, $f = 11$ and $\alpha_t = 0.001$.

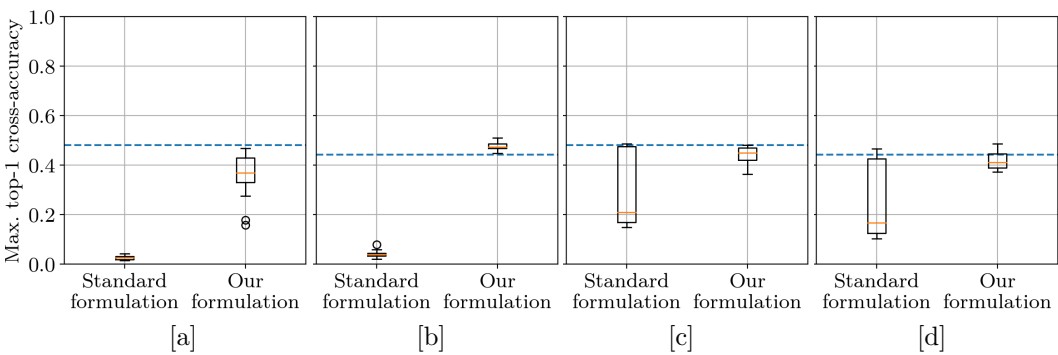

Figure 15: CIFAR-100 and *convolutional* model, with $n = 25$, $f = 5$ and $\alpha_t = 0.01$ if $t < 1500$ else $\alpha_t = 0.001$.

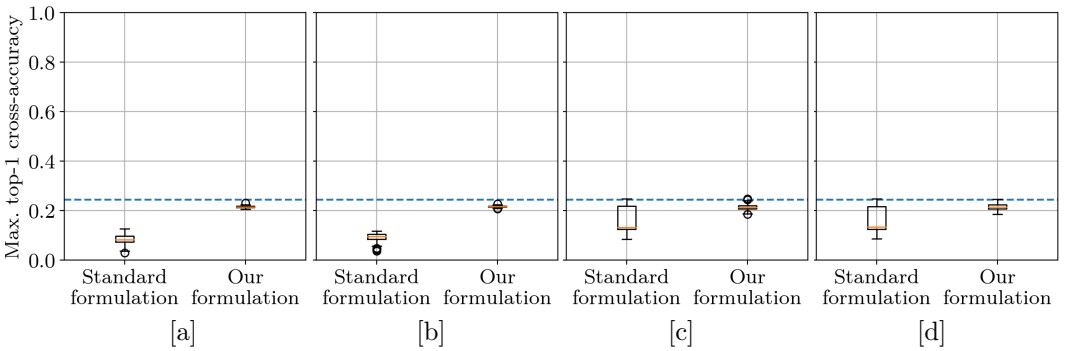

Figure 16: CIFAR-100 and *convolutional* model, with $n = 25$, $f = 5$ and $\alpha_t = 0.001$.

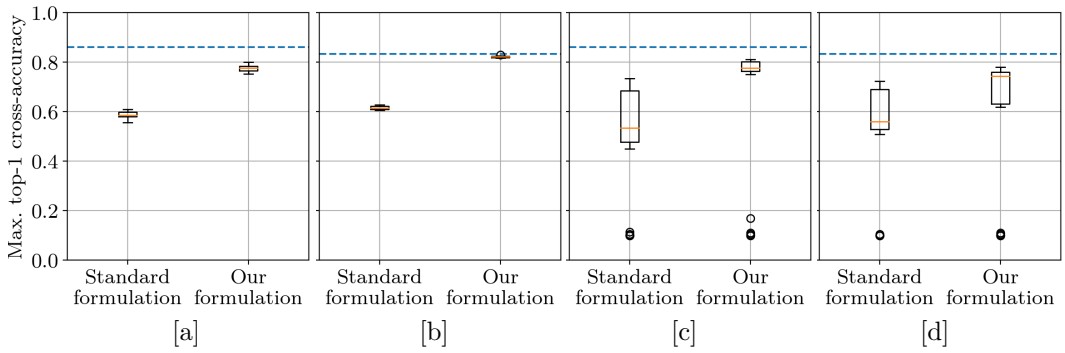

Figure 17: Fashion MNIST and *fully connected* model, with $n = 51$, $f = 24$ and $\alpha_t = 0.5$.

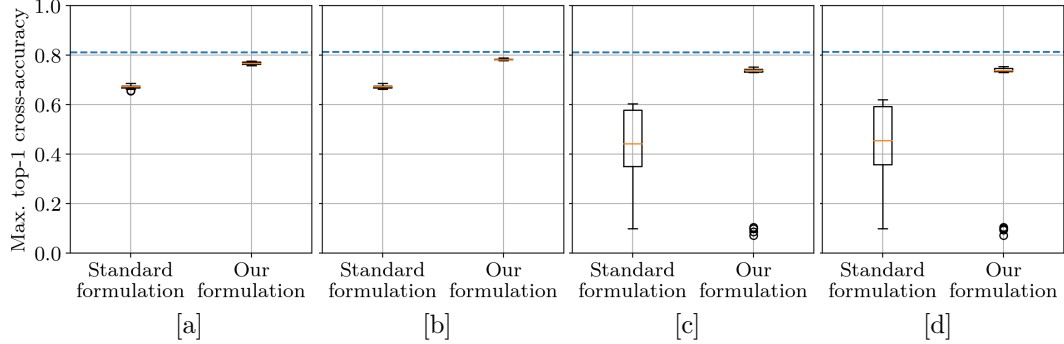

Figure 18: Fashion MNIST and *fully connected* model, with $n = 51$, $f = 24$ and $\alpha_t = 0.02$.

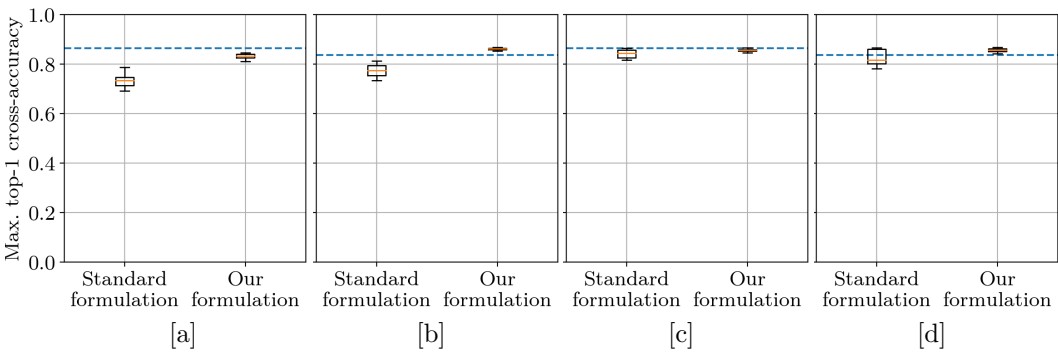

Figure 19: Fashion MNIST and *fully connected* model, with $n = 51$, $f = 12$ and $\alpha_t = 0.5$.

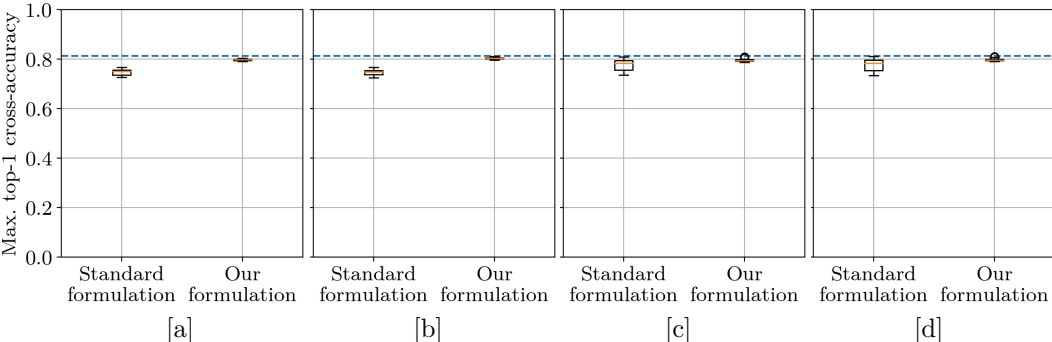

Figure 20: Fashion MNIST and *fully connected* model, with $n = 51$, $f = 12$ and $\alpha_t = 0.02$.

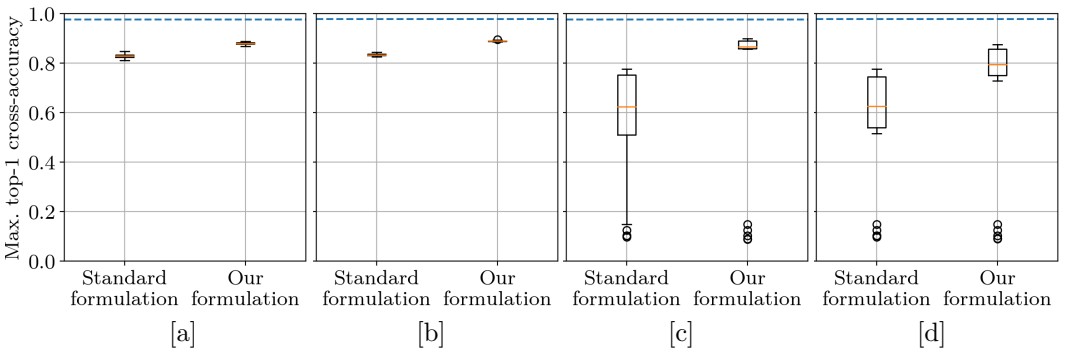

Figure 21: MNIST and *fully connected* model, with $n = 51$, $f = 24$ and $\alpha_t = 0.5$.

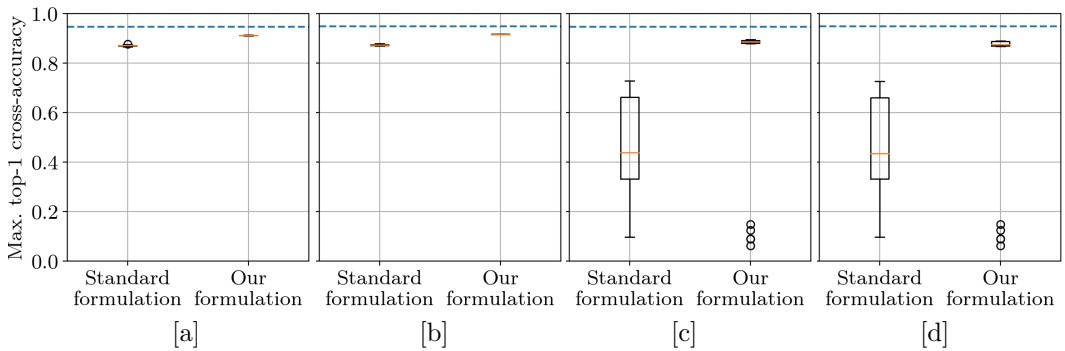

Figure 22: MNIST and *fully connected* model, with $n = 51$, $f = 24$ and $\alpha_t = 0.02$.

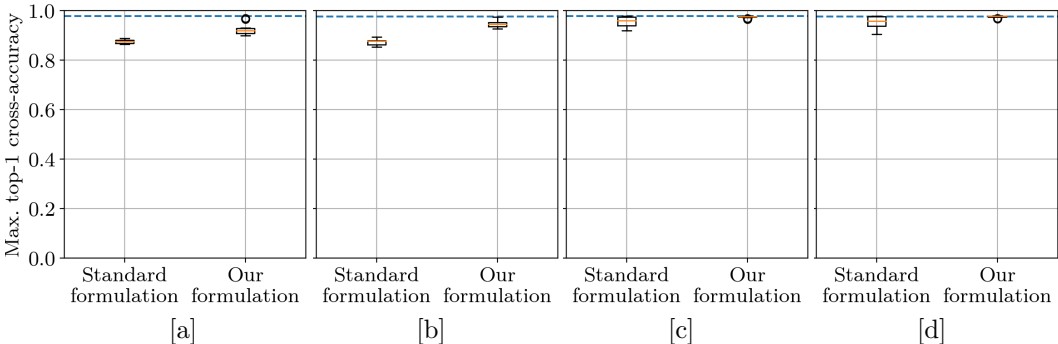

Figure 23: MNIST and *fully connected* model, with $n = 51$, $f = 12$ and $\alpha_t = 0.5$.

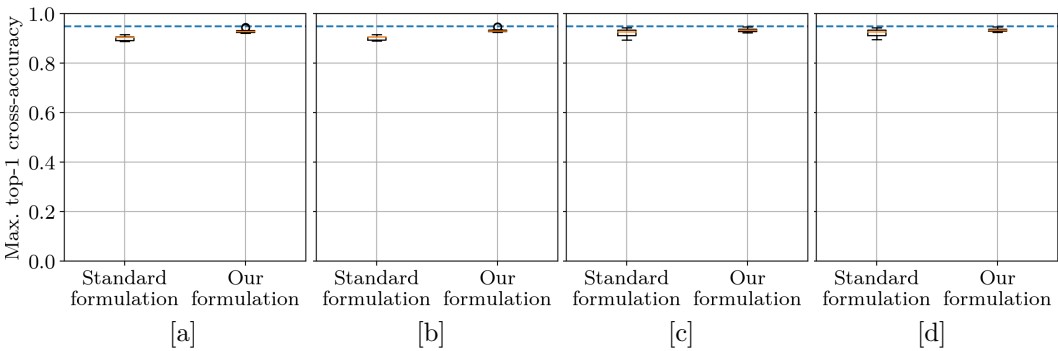

Figure 24: MNIST and *fully connected* model, with $n = 51$, $f = 12$ and $\alpha_t = 0.02$.

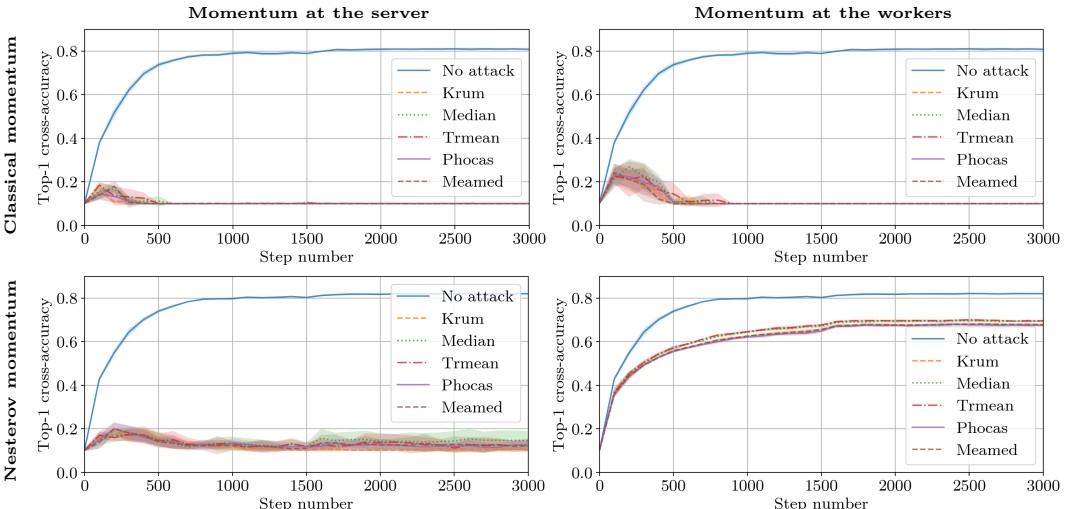

Figure 25: CIFAR-10 dataset and *convolutional* model, with $n = 25$, $f = 11$ and $\alpha_t = 0.01$ if $t < 1500$ else $\alpha_t = 0.001$, under attack from Baruch et al. (2019).

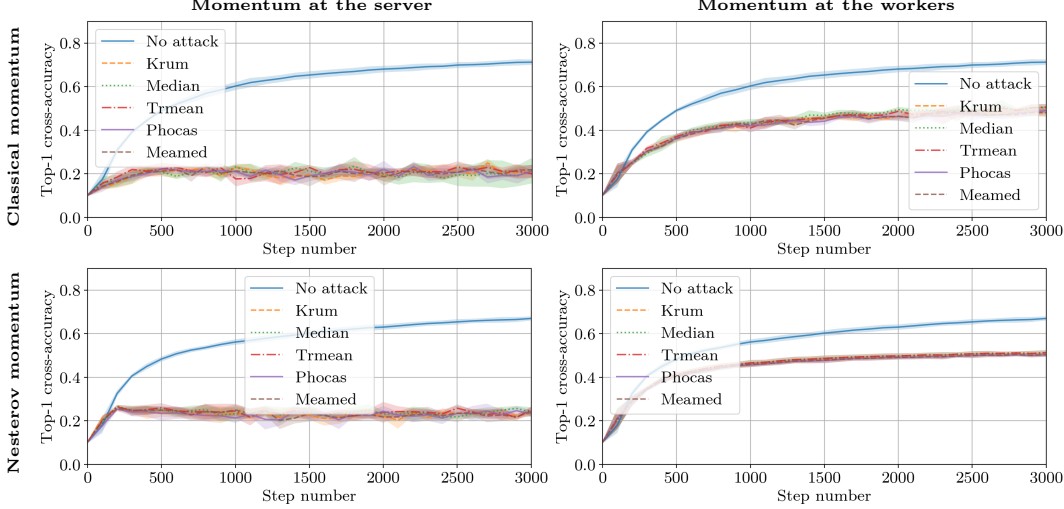

Figure 26: CIFAR-10 dataset and *convolutional* model, with $n = 25$, $f = 11$ and $\alpha_t = 0.001$, under attack from Baruch et al. (2019).

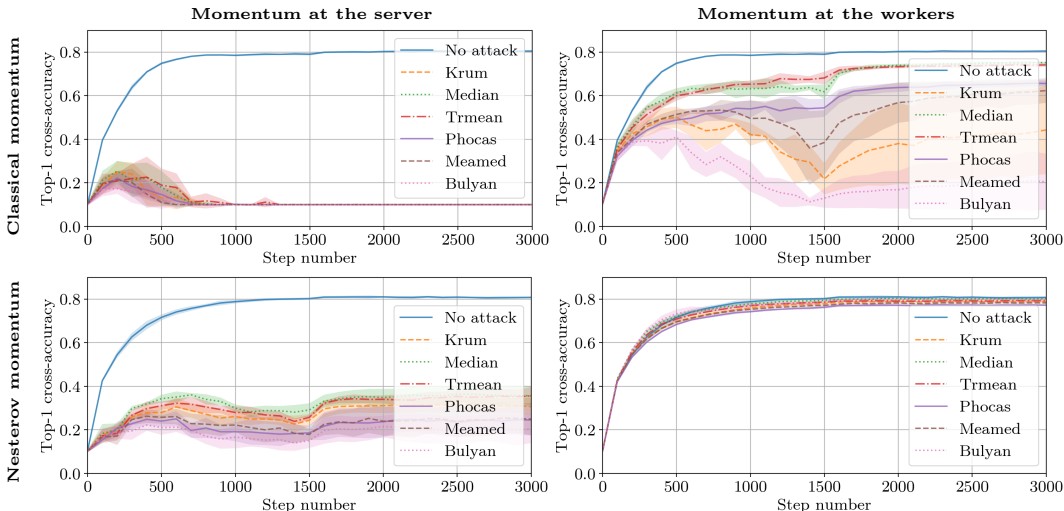

Figure 27: CIFAR-10 dataset and *convolutional* model, with $n = 25$, $f = 5$ and $\alpha_t = 0.01$ if $t < 1500$ else $\alpha_t = 0.001$, under attack from Baruch et al. (2019).

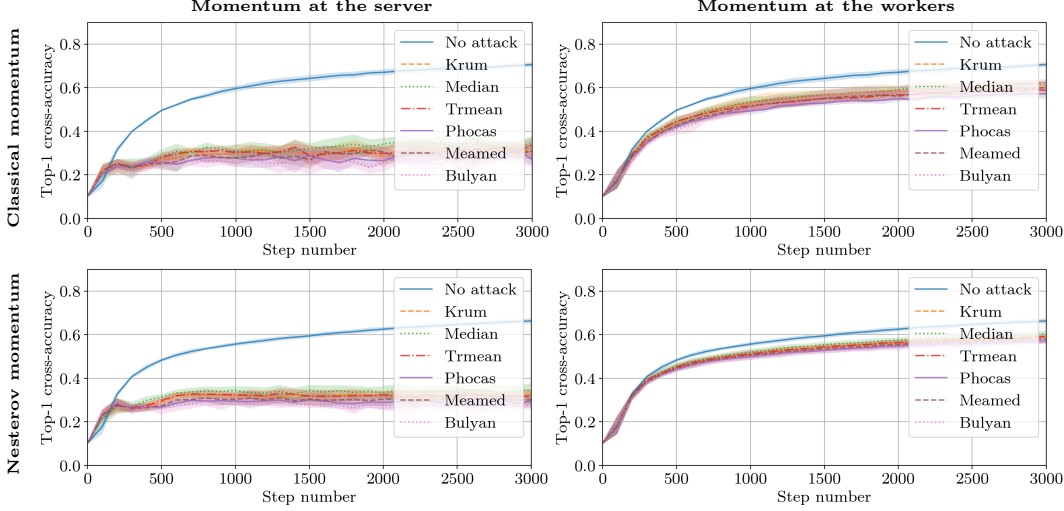

Figure 28: CIFAR-10 dataset and *convolutional* model, with $n = 25$, $f = 5$ and $\alpha_t = 0.001$, under attack from Baruch et al. (2019).

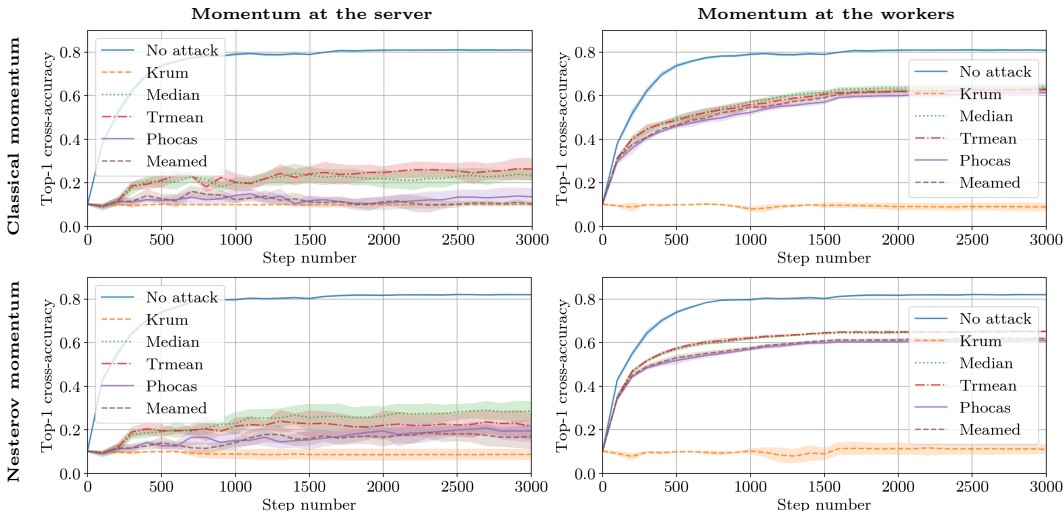

Figure 29: CIFAR-10 dataset and *convolutional* model, with $n = 25$, $f = 11$ and $\alpha_t = 0.01$ if $t < 1500$ else $\alpha_t = 0.001$, under attack from Xie et al. (2019a).

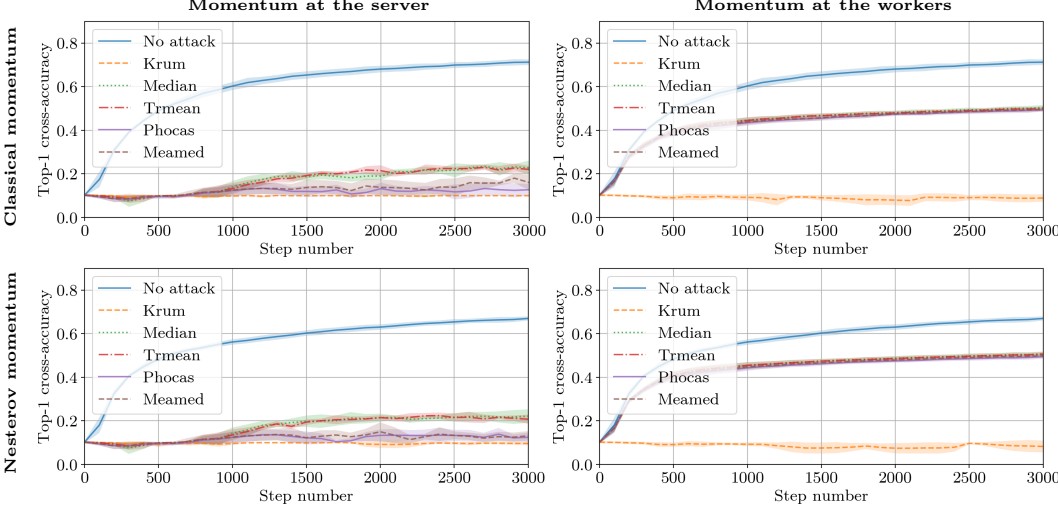

Figure 30: CIFAR-10 dataset and *convolutional* model, with $n = 25$, $f = 11$ and $\alpha_t = 0.001$, under attack from Xie et al. (2019a).

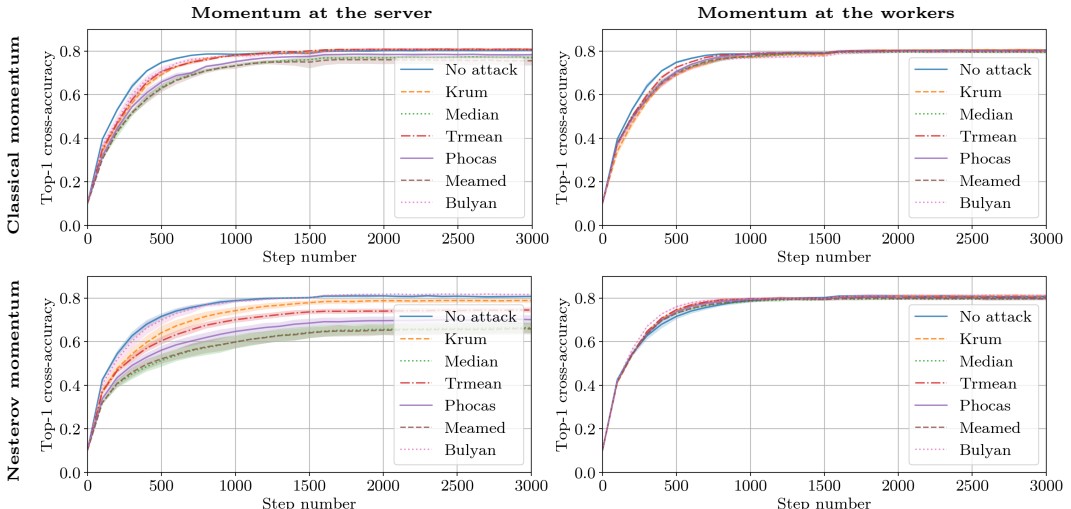

Figure 31: CIFAR-10 dataset and *convolutional* model, with $n = 25$, $f = 5$ and $\alpha_t = 0.01$ if $t < 1500$ else $\alpha_t = 0.001$, under attack from Xie et al. (2019a).

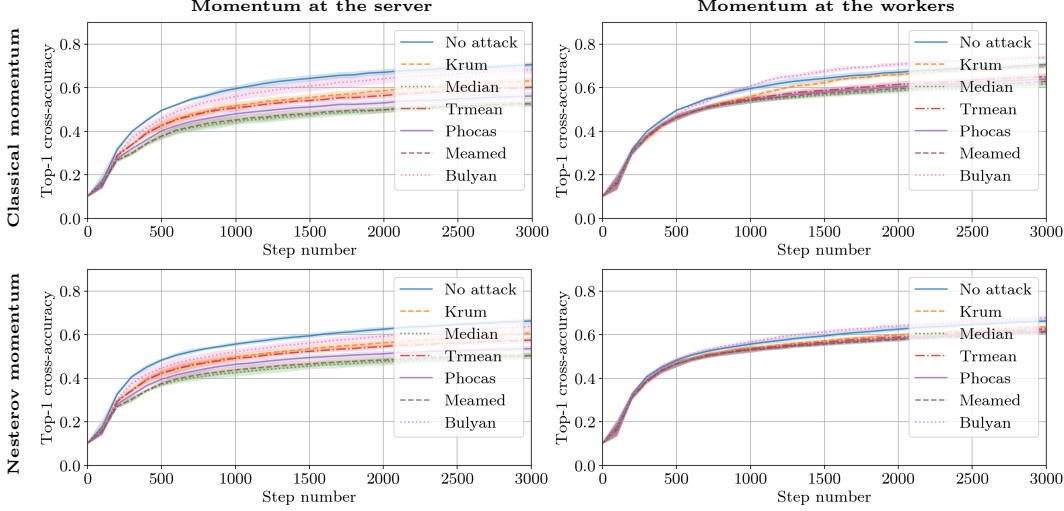

Figure 32: CIFAR-10 dataset and *convolutional* model, with $n = 25$, $f = 5$ and $\alpha_t = 0.001$, under attack from Xie et al. (2019a).

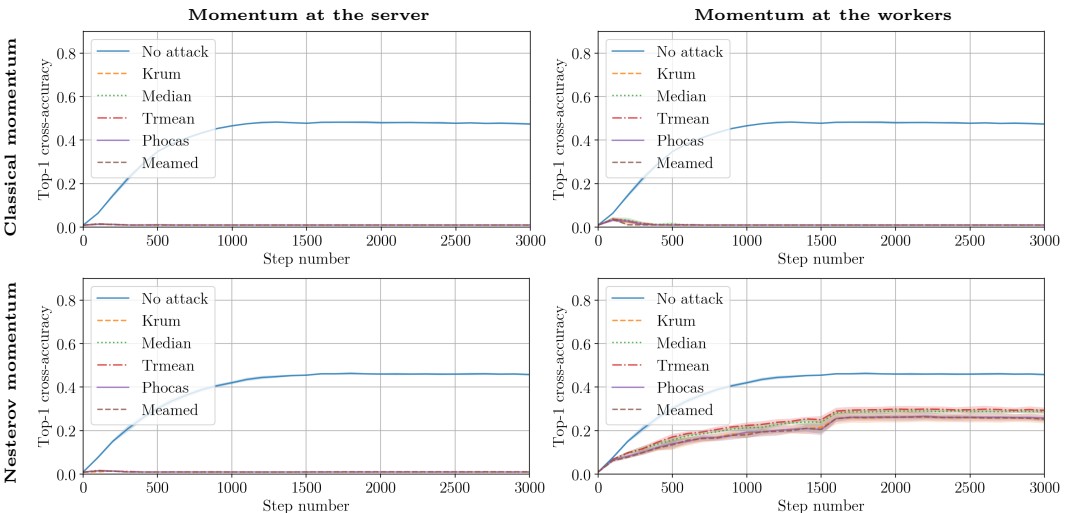

Figure 33: CIFAR-100 dataset and *convolutional* model, with $n = 25$, $f = 11$ and $\alpha_t = 0.01$ if $t < 1500$ else $\alpha_t = 0.001$, under attack from Baruch et al. (2019).

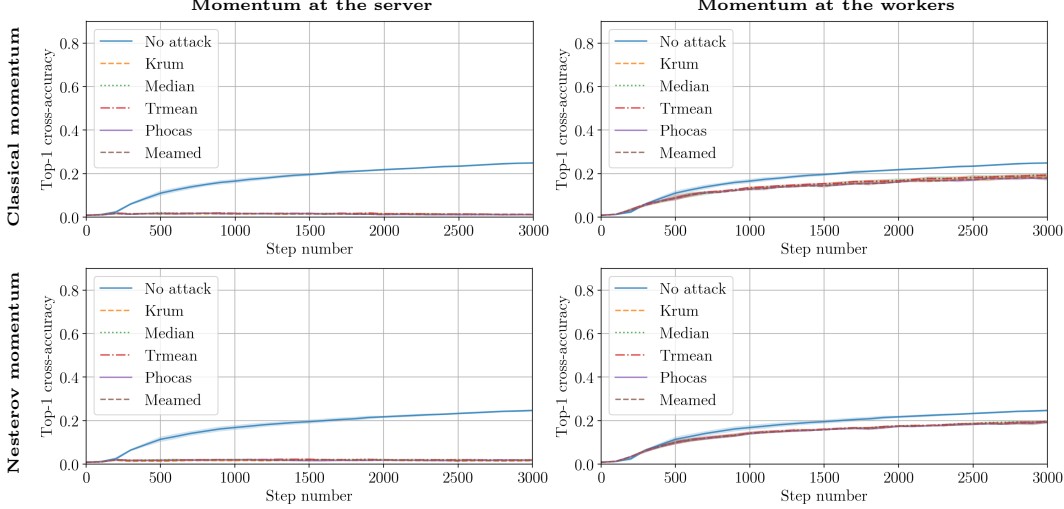

Figure 34: CIFAR-100 dataset and *convolutional* model, with $n = 25$, $f = 11$ and $\alpha_t = 0.001$, under attack from Baruch et al. (2019).

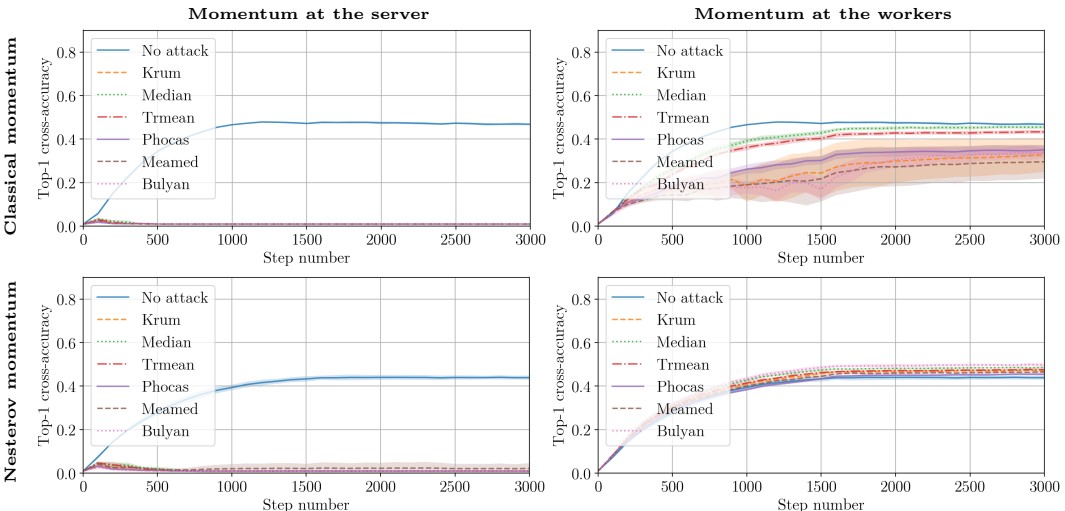

Figure 35: CIFAR-100 dataset and *convolutional* model, with $n = 25$, $f = 5$ and $\alpha_t = 0.01$ if $t < 1500$ else $\alpha_t = 0.001$, under attack from Baruch et al. (2019).

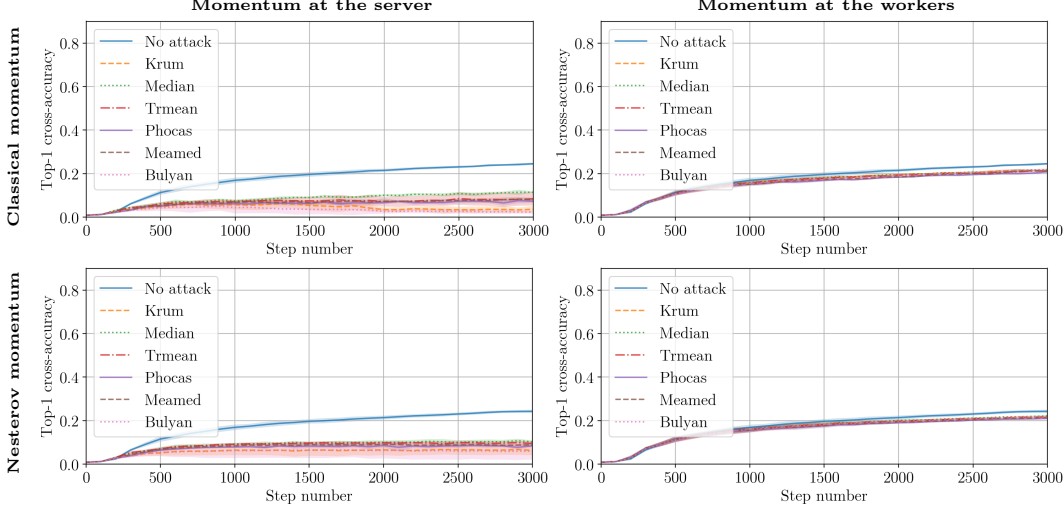

Figure 36: CIFAR-100 dataset and *convolutional* model, with $n = 25$, $f = 5$ and $\alpha_t = 0.001$, under attack from Baruch et al. (2019).

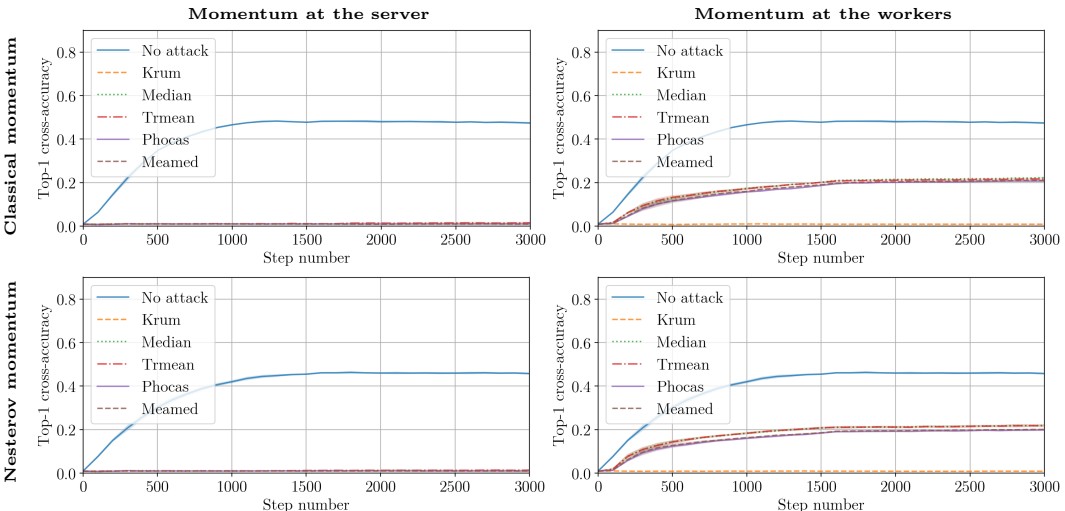

Figure 37: CIFAR-100 dataset and *convolutional* model, with $n = 25$, $f = 11$ and $\alpha_t = 0.01$ if $t < 1500$ else $\alpha_t = 0.001$, under attack from Xie et al. (2019a).

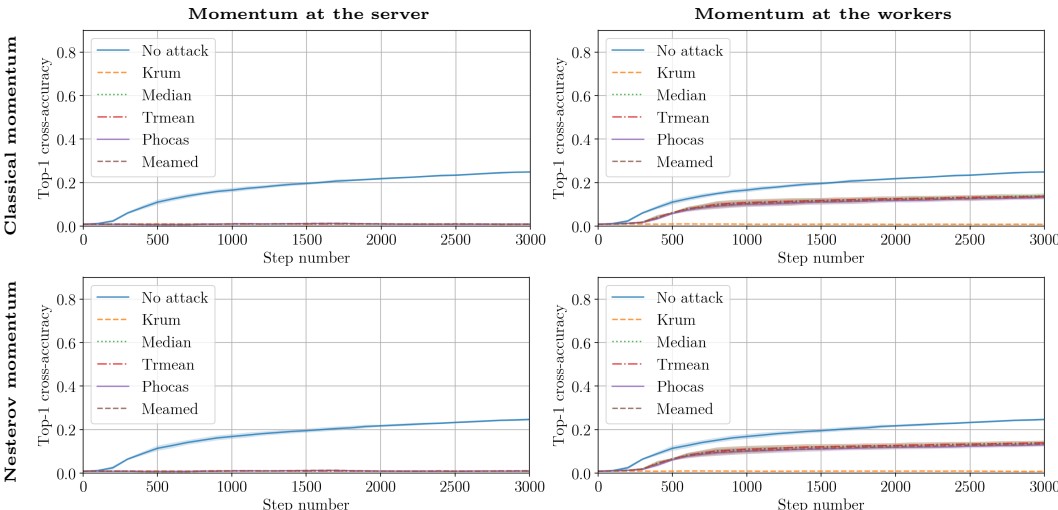

Figure 38: CIFAR-100 dataset and *convolutional* model, with $n = 25$, $f = 11$ and $\alpha_t = 0.001$, under attack from Xie et al. (2019a).

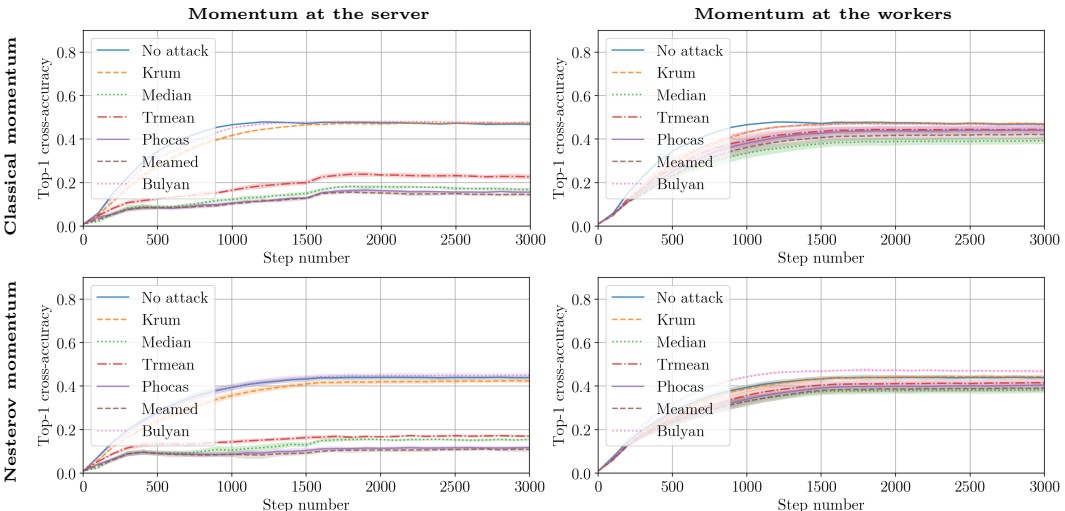

Figure 39: CIFAR-100 dataset and *convolutional* model, with $n = 25$, $f = 5$ and $\alpha_t = 0.01$ if $t < 1500$ else $\alpha_t = 0.001$, under attack from Xie et al. (2019a).

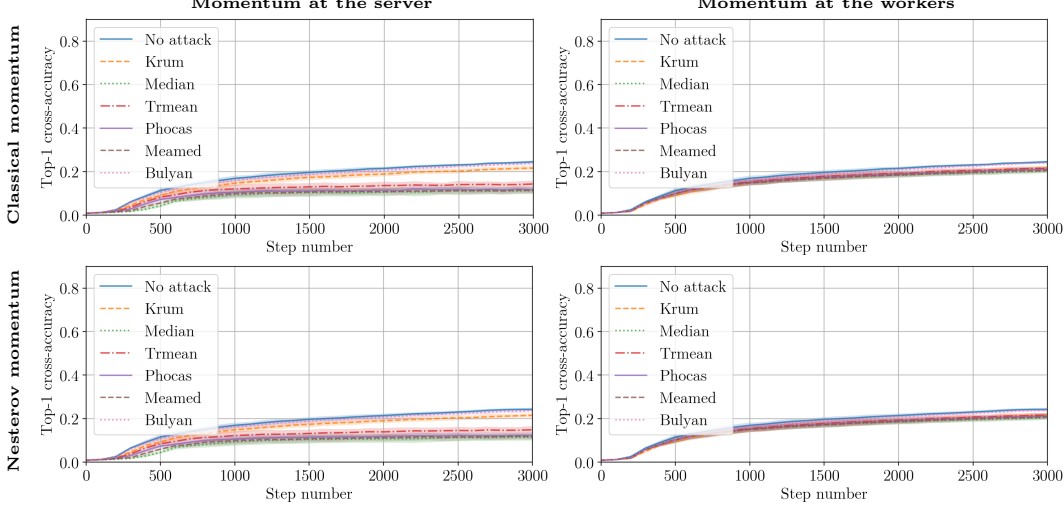

Figure 40: CIFAR-100 dataset and *convolutional* model, with $n = 25$, $f = 5$ and $\alpha_t = 0.001$, under attack from Xie et al. (2019a).

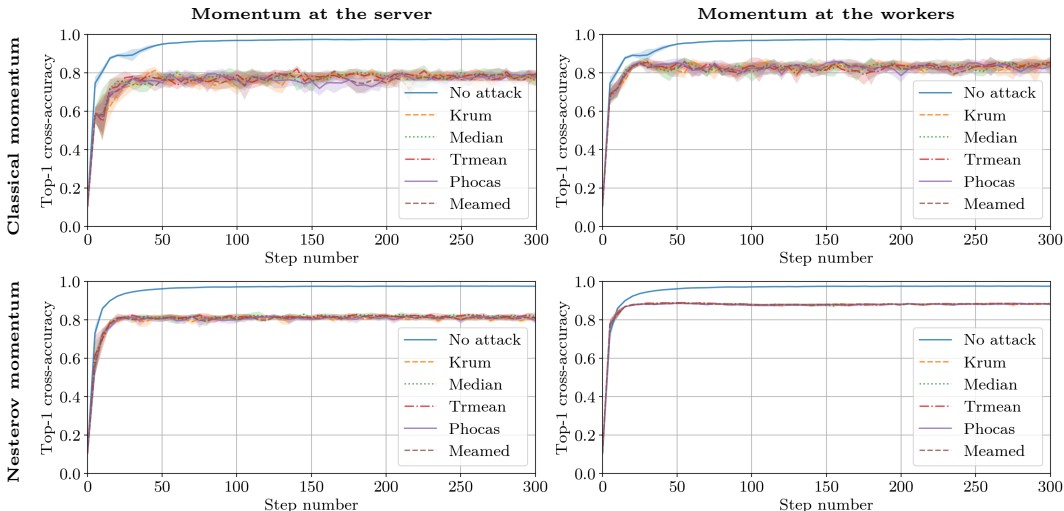

Figure 41: MNIST dataset and *fully connected* model, with $n = 51$, $f = 24$ and $\alpha_t = 0.5$, under attack from Baruch et al. (2019).

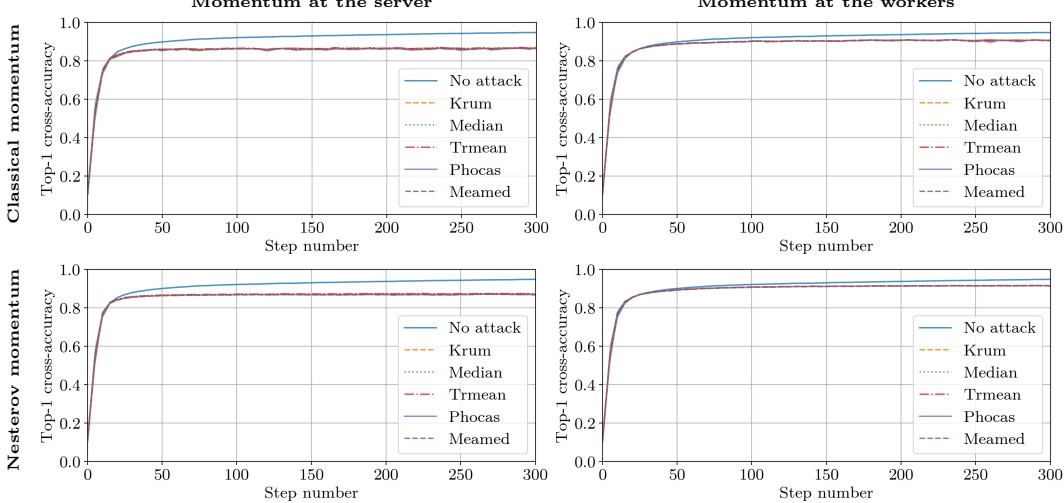

Figure 42: MNIST dataset and *fully connected* model, with $n = 51$, $f = 24$ and $\alpha_t = 0.02$, under attack from Baruch et al. (2019).

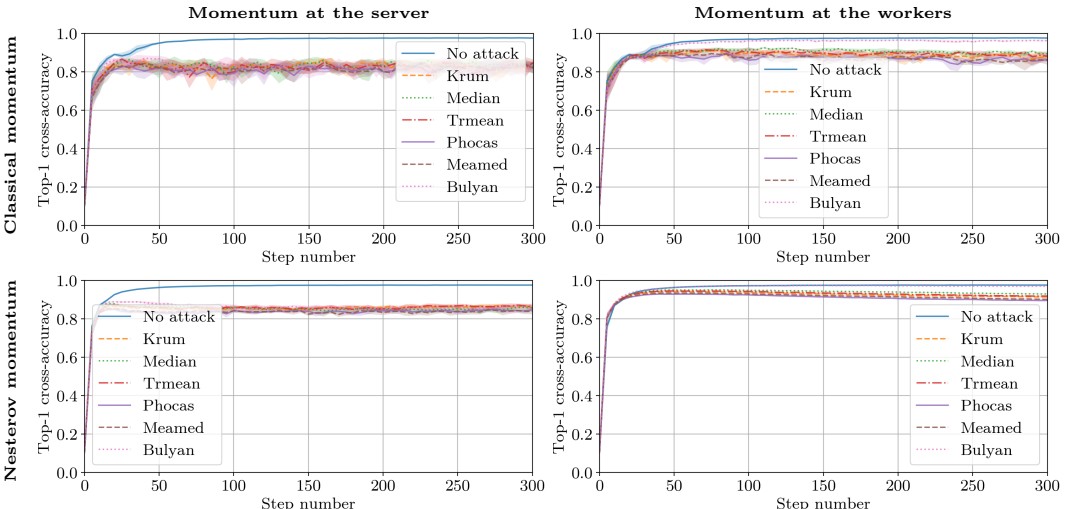

Figure 43: MNIST dataset and *fully connected* model, with $n = 51$, $f = 12$ and $\alpha_t = 0.5$, under attack from Baruch et al. (2019).

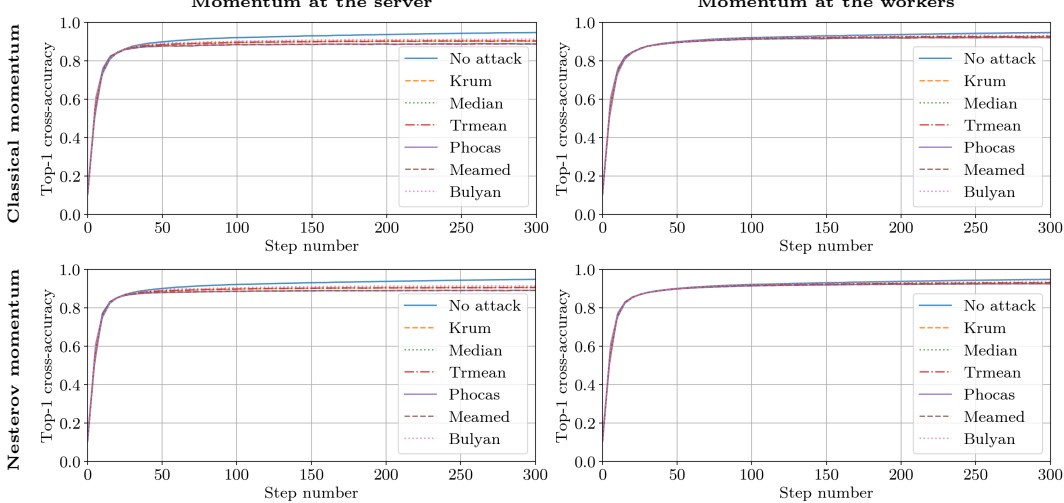

Figure 44: MNIST dataset and *fully connected* model, with $n = 51$, $f = 12$ and $\alpha_t = 0.02$, under attack from Baruch et al. (2019).

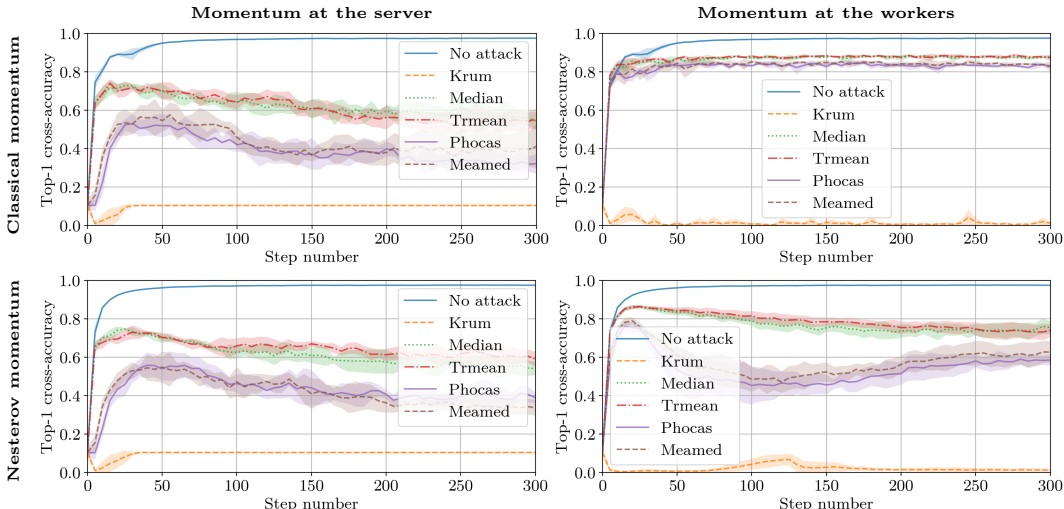

Figure 45: MNIST dataset and *fully connected* model, with $n = 51$, $f = 24$ and $\alpha_t = 0.5$, under attack from Xie et al. (2019a).

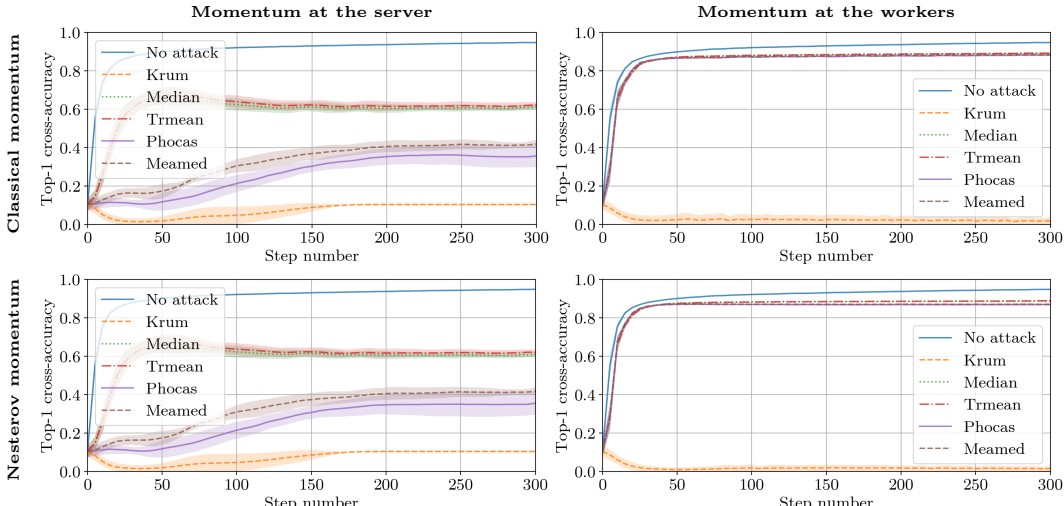

Figure 46: MNIST dataset and *fully connected* model, with $n = 51$, $f = 24$ and $\alpha_t = 0.02$, under attack from Xie et al. (2019a).

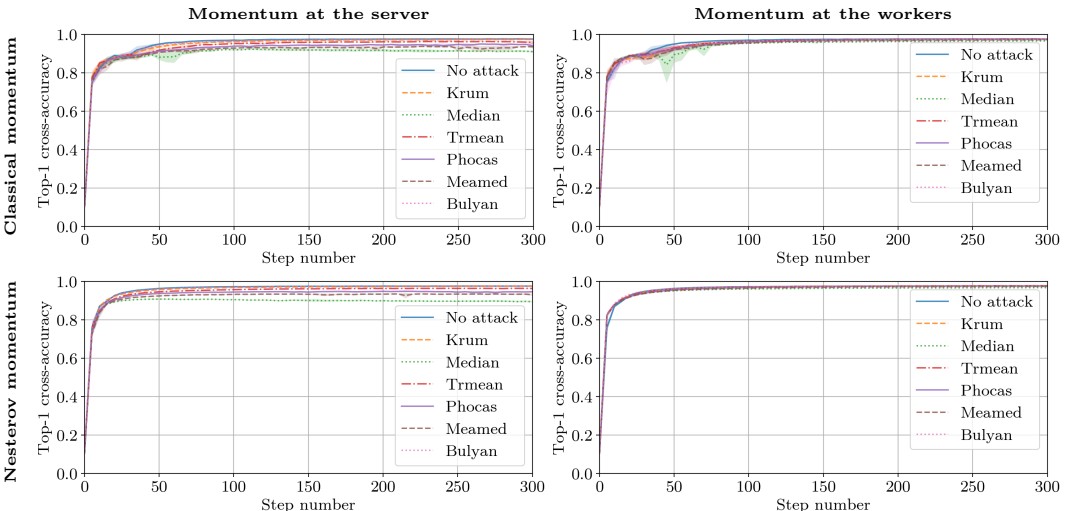

Figure 47: MNIST dataset and *fully connected* model, with $n = 51$, $f = 12$ and $\alpha_t = 0.5$, under attack from Xie et al. (2019a).

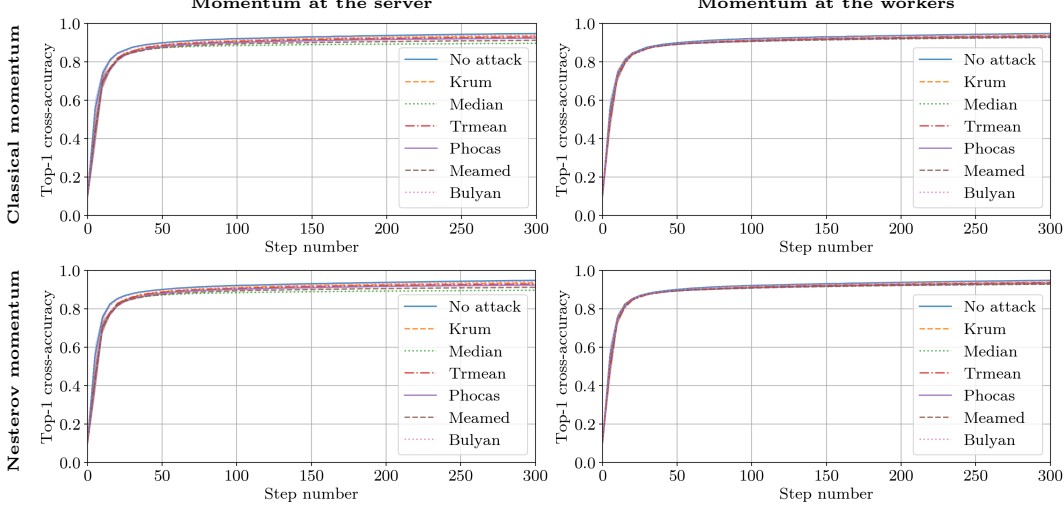

Figure 48: MNIST dataset and *fully connected* model, with $n = 51$, $f = 12$ and $\alpha_t = 0.02$, under attack from Xie et al. (2019a).

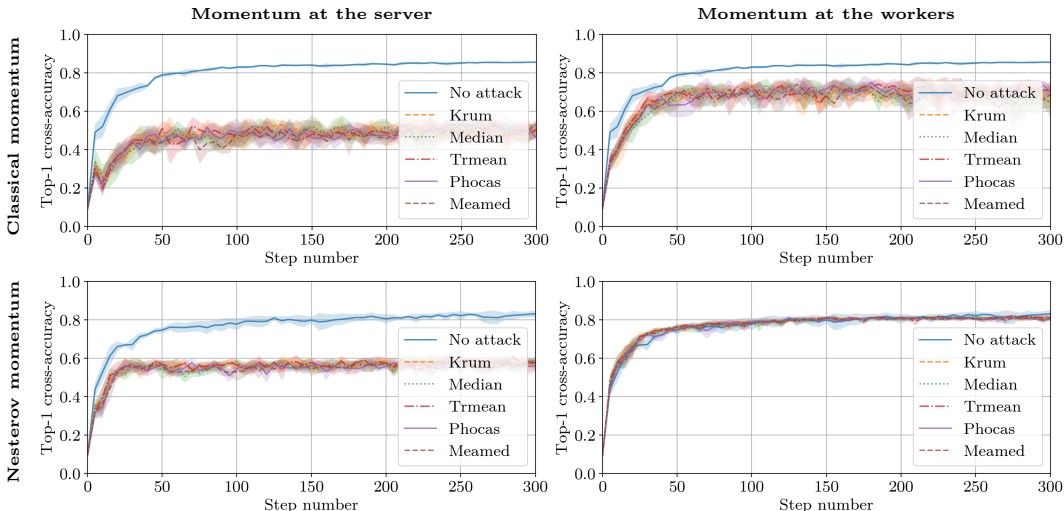

Figure 49: Fashion MNIST dataset and *fully connected* model, with $n = 51$, $f = 24$ and $\alpha_t = 0.5$, under attack from Baruch et al. (2019).

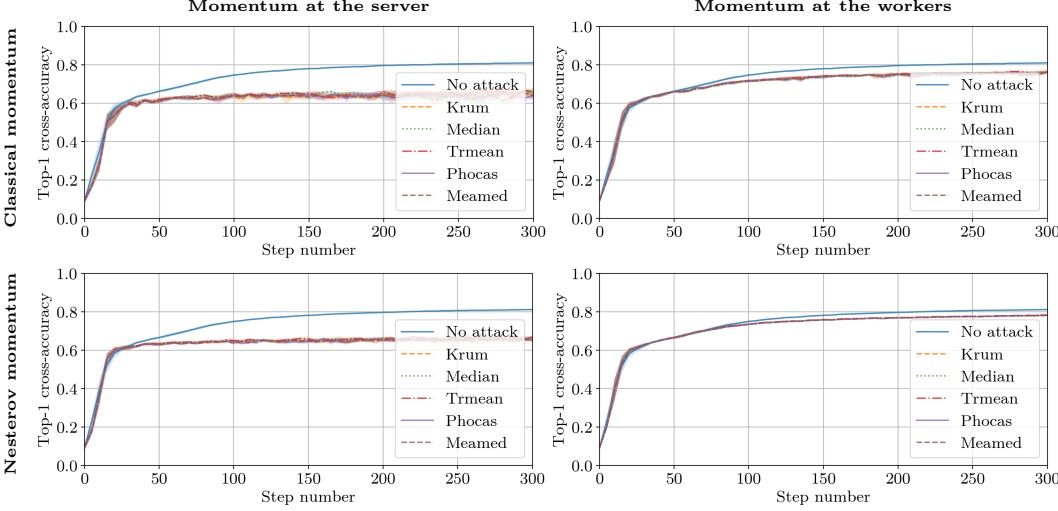

Figure 50: Fashion MNIST dataset and *fully connected* model, with $n = 51$, $f = 24$ and $\alpha_t = 0.02$, under attack from Baruch et al. (2019).

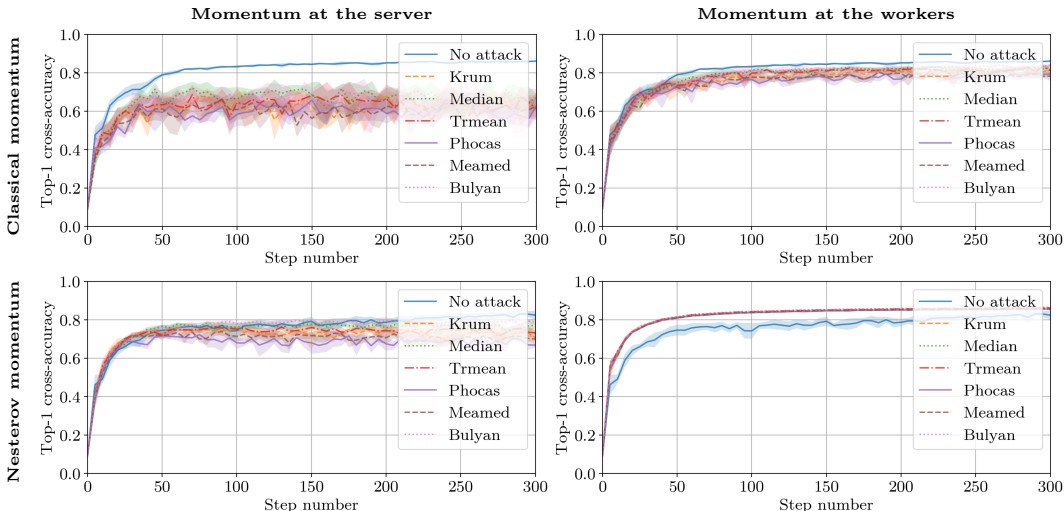

Figure 51: Fashion MNIST dataset and *fully connected* model, with $n = 51$, $f = 12$ and $\alpha_t = 0.5$, under attack from Baruch et al. (2019).

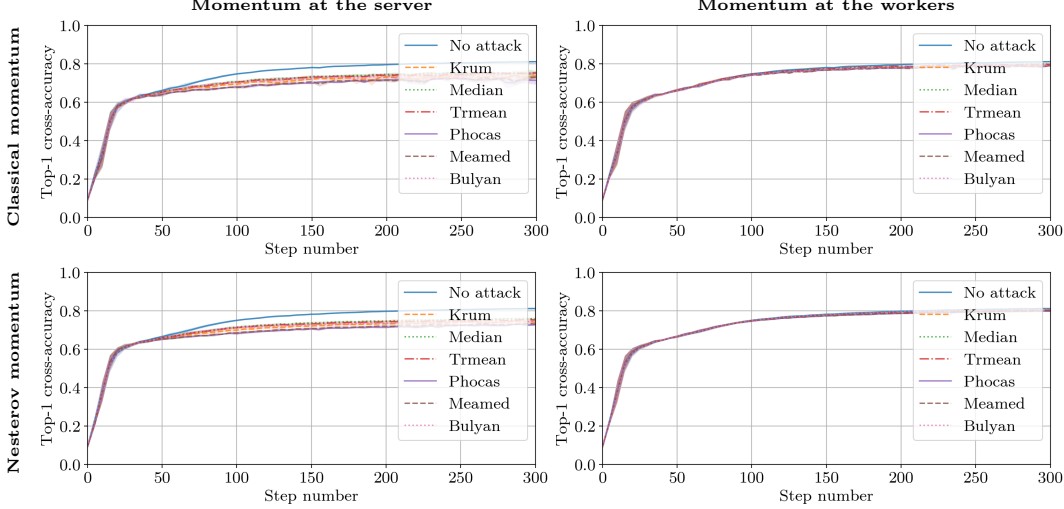

Figure 52: Fashion MNIST dataset and *fully connected* model, with $n = 51$, $f = 12$ and $\alpha_t = 0.02$, under attack from Baruch et al. (2019).

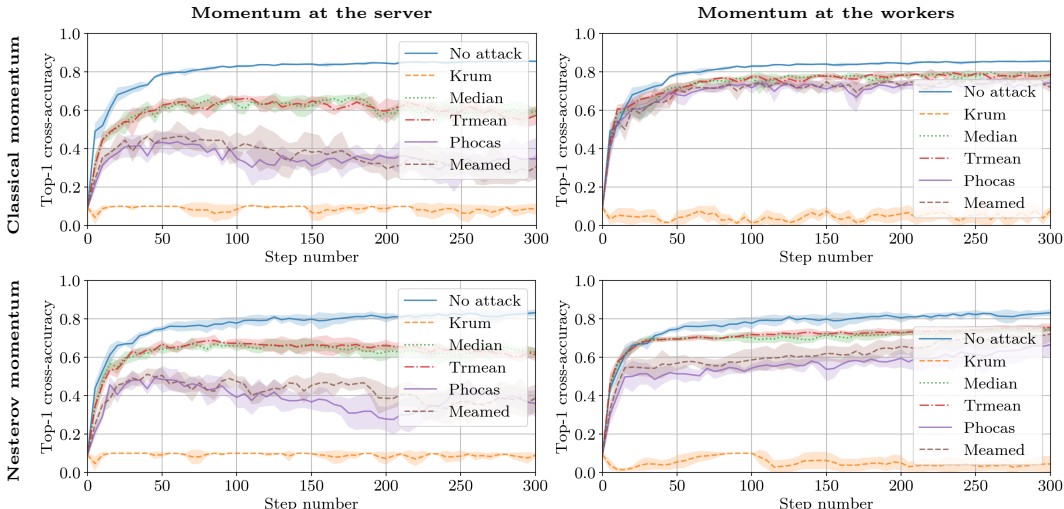

Figure 53: Fashion MNIST dataset and *fully connected* model, with $n = 51$, $f = 24$ and $\alpha_t = 0.5$, under attack from Xie et al. (2019a).

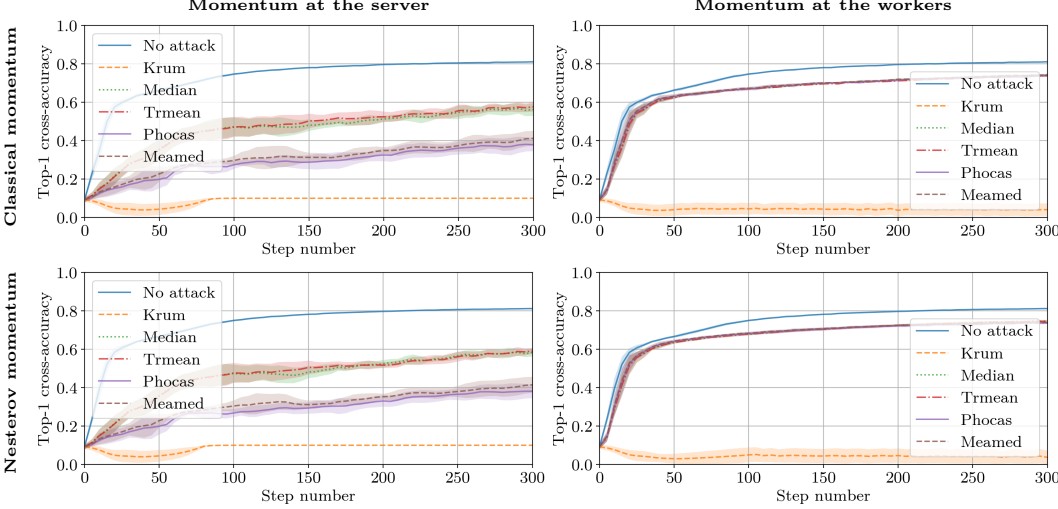

Figure 54: Fashion MNIST dataset and *fully connected* model, with $n = 51$, $f = 24$ and $\alpha_t = 0.02$, under attack from Xie et al. (2019a).

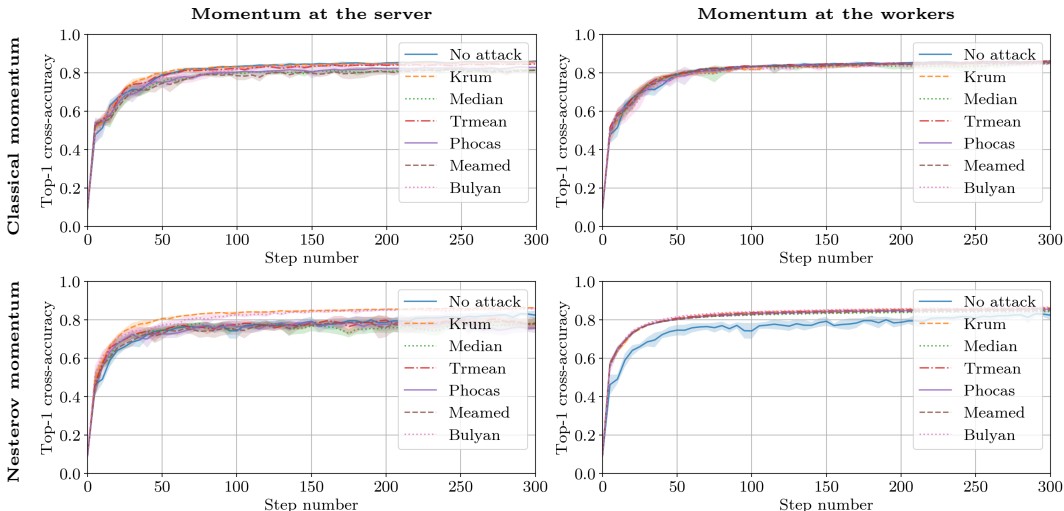

Figure 55: Fashion MNIST dataset and *fully connected* model, with $n = 51$, $f = 12$ and $\alpha_t = 0.5$, under attack from Xie et al. (2019a).

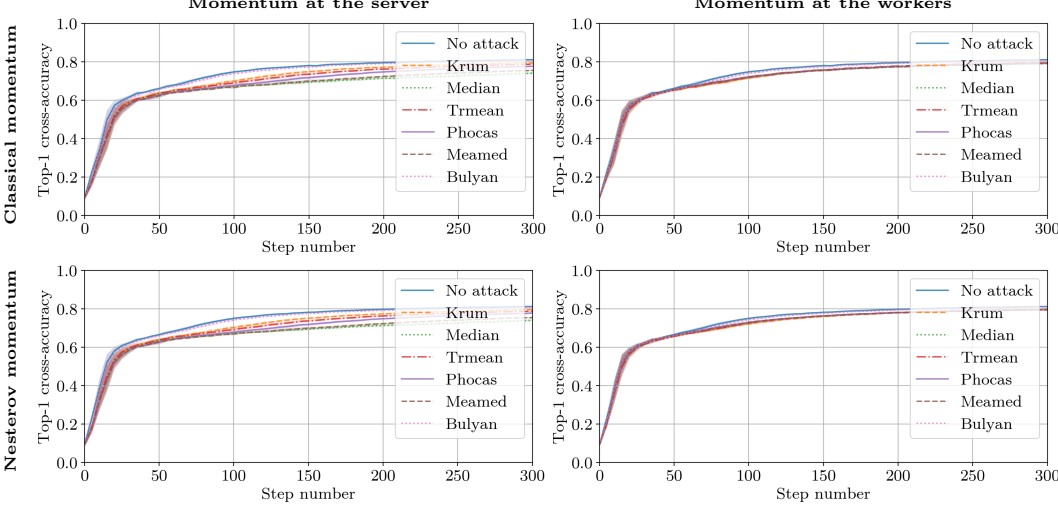

Figure 56: Fashion MNIST dataset and *fully connected* model, with $n = 51$, $f = 12$ and $\alpha_t = 0.02$, under attack from Xie et al. (2019a).

