# OpenReview forum: "Distributed Momentum for Byzantine-resilient Stochastic Gradient Descent"
_ICLR.cc/2021/Conference — ICLR 2021 Poster_

### Official Review · AnonReviewer3 · 2020-10-28
**Hard to get in the way of something with momentum**

**Rating:** 6
**Confidence:** 2

**Review:**

Summary:
The paper describes an approach to counteract Byzantine attacks for distributed stochastic gradient descent by using the momentum of the gradient computed at the workers, which relies only on memory of the previous momentum. This seems to thwart current attacks in the majority of scenarios tested. The theoretical analysis seems appropriate. The empirical results are accompanied by precise details and should facilitate reproducibility (given a week of computation time).

Strengths:
The paper is well organized. The proposed approach addresses known attacks. The results are done in a rigorous and reproducible manner. Related work is highlighted.

Weaknesses:
The text in the introduction is a bit overreaching. Perhaps, "main driving force behind the successes of machine learning" replaced by "main optimization algorithm used throughout machine learning". Also, is not clear that data that are not "well-sanitized" and the existence of "software bugs" could be considered colluding adversaries. These situations certainly do not constitute omniscient adversaries.  Their inclusion does not seem within the scenarios considered.  In this vein, I don't find much realism of the addressed attack scenarios.

The nature of the failure mechanism for cases when the defense is not successful could be made more clear or investigated more.

The related and future work section were helpful, but the text in that section are too terse, especially when new concepts are introduced. More text describing these concepts and relating them to the situations addressed by the paper would be helpful.

Conclusion:
I think the paper is  well written contribution to the literature on Byzantine attacks for stochastic gradient descent.  I think it will be significant to researchers in the area. However, it is outside my expertise and I found interpreting the contribution difficult.

Other suggestions:
Figure 5 is difficult to understand without a contrast of what it would be without the proposed momentum at the workers.

Section 5 could be expanded. It is not clear why the "suspicious-based" fault tolerance is considered a more appropriate defense. Could the proposed momentum at worker approach make it easier to detect attacks (rather than defend)?

The detailed results of learning curves in the Appendix are easy to grasp, but the number of figures seems excessive versus summarizing the loss at a certain iteration or the number of iterations to a certain performance level.

Minor:
The font seems different as compared to other submissions.

In the abstract, "with seeds 1 to 5" -> "with specified seeds (1 to 5)"

In section 3.1 the definition and then redefinition in equation 2 is a bit confusing. Why not differentiate the notation?

References to books titles like Bottou's should be capitalized.

Page 14  "Basically, and" -> "Basically,"

---

> ### Author Response · Authors · 2020-11-17
> **Response: Hard to get in the way of something with momentum**
>
> We thank the reviewer for their detailed review.
>
> (We would like to add that, beside precise experimental details, our code is also distributed (along with a script reproducing all the experiments and graphs presented in this paper) to further facilitate reproducibility/in-depth assessment of our experiments.)
>
> The reformulation you propose is indeed more factual/less debatable. We adopted it.
>
> Ill-sanitized data would indeed not be considered a colluding adversary, but "software bugs" can offer an adversary the opportunity to take control of vulnerable nodes in actual deployments. (We replaced "software bugs" by "software vulnerabilities" in the paper.) The compromised nodes, under the control of the same adversary, could then collude.
>
> The threat model is inherited from the threat model of the defenses, which all claim to thwart omniscient adversaries.
> The realism of the considered attacks can remain a subject for debate (Baruch et al. (2019) defended that they could be implemented in practice), but we believe most of their value to the community was anyway to show that the assumptions behind the resilience proofs were actually unverified in practice (c.f. Baruch et al. (2019), Section 4.1).
>
> Identifying the precise nature of the failure mechanism impacting these defenses, especially when the model is as complex as a neural network (with at the very least hundreds of thousand of parameters), would represent a substantial step forward in the topic of Byzantine SGD. While we do not claim our results allow to precisely identify all the causes of failure, we argue they offer a solid empirical evidence that the variance-norm ratio of the honest gradients is one important metric to predict the effectiveness of statistically-robust defenses.
>
> As we are allowed one more page, we have used it in part to expand the future work section. We have also provided, at the beginning of Section D (now Section E), summaries of the maximum observed top-1 cross-accuracy (in the same format as in Figure 1) for all our experiments. We believe keeping the learning curves (in the first version of the paper, figures 6 to 37) right after would remain a nice addition.
>
> Figure 5 reports on the measured variance-norm ratio in the same settings as in Figure 4. It is the experimental counterpart of the theoretical analysis led in Section B.2: this figure actually compares the quantities $r_t^{(s)}$ and $r_t^{(w)}$ at every step $t$. In particular it empirically confirms Equation 8, that lowering the learning rate would decrease the variance-norm ratio (the "dip" right after step $t = 1500$). This observation could be used in a future work, to build an adaptive learning rate which decreases depending on the curvature of the parameter trajectory (the quantity $s_t$, which can also be measured while training).
>
> Perhaps there is a misunderstanding, as we do not consider the "suspicion-based" scheme a more appropriate defense.
> We have extended our discussion in Section 5.
>
> To precisely answer your question (whether it would make it _easier_), yes, our method is going in the right direction by reducing the variance-norm ratio (e.g. take the extreme case of a null ratio, then every honest gradient would be equal and an attack would be detected if its gradient differs from the honest majority).
> We nevertheless believe that (reliably) detecting attacks is a tougher challenge than filtering potential attacks.
> Note that even low-bias-low-variance "suspicion-based" schemes would not be sufficient either to reliably detect an attack: an attack could both decrease the loss and embed adversarial behaviors, especially as the target model may often be over-parameterized.
>
> The font is indeed slightly different (the size is good but the glyphs are slightly different). We have re-downloaded and re-uploaded (in our Overleaf project) the official ICLR21 template, cleared the cache and re-compiled the project, without any change.
> We are working on it.
>
> We perfectly understand that the redefinition of $G_t$ may be surprising to some, but we believe this is a trade-off. Introducing yet another notation (with which e.g. the update equation would not be valid anymore) may be more confusing to some other than a redefinition.
>
> The other minor comments have been addressed in the new version.

---

### Official Review · AnonReviewer2 · 2020-10-28
**Paper is well written but need more formal and statistical analysis**

**Rating:** 4
**Confidence:** 3

**Review:**

The paper's method is quite simple and it argues that the latest distributed stochastic gradient descent (SGD) state-of-the- based on the Byzantine model can be tackled using momentum-based versions of (SGD). There is a slight issue with the Nesterov variant that the authors are using. I don't know they have used the formulation to calculate gradients at $\theta_t-\alpha v_t$ instead of calculating gradients at $\theta_t-\mu v_t$, where $\mu$ is called momentum. Other than that they have used the aggregating functions which are previously studied and known to $(\alpha, f)$-resilient. There is not much contribution by the authors in this area. Although they studied these aggregation rules under recently presented attacks.
In supplementary material, authors show that computing momentum-based gradient at workers under some assumptions they can keep the variance-norm ratio low but they already have discussed that this might be an issue as standard deviation can be negative. More insight and analysis are required and I am not sure how it may behave other attacks that are not studied in this paper. The results are quite detailed but due to the huge number of combinations, it is difficult to summarize all settings, and the numbers already stated are not in favor. When half of the machines are Byzantine 20\% accuracy is recovered in 49.25\% cases which is not a favorable number. The number of stats for results are missing like average recovered accuracy, median and basic analysis charts could be more useful than adding charts of different combinations of possible experiments.

---

> ### Author Response · Authors · 2020-11-17
> **Response: Paper is well written but need more formal and statistical analysis**
>
> We thank the reviewer for their review.
>
> There was indeed a typo in the paper, but the gradient is actually to be estimated at $\theta_t - \alpha_t \mu v_t$ to follow the formulation of Nesterov's accelerated gradient. Please note that the code was correctly implemented (see lines 717 and 722 in attack.py). The two update equations we wrote are the same as the ones in e.g. [1] (equations 3 and 4), except for one change of variable (so that the update equation remains mostly unchanged, namely: $\theta_{t + 1} = \theta_t - \alpha_t v_{t + 1}$ compared to $\theta_{t + 1} = \theta_t - \alpha_t g_t$ without momentum).
>
> We would like to offer some perspective on the work presented in this paper.
> To the best of our knowledge, the relevant literature is currently composed of several statistically-robust defenses and three attacks, which all follow the same core principle (see Section 2.3). This core principle may appear rather "ad hoc" (we will not disagree with AnonReviewer1 on this point), but so little is actually enough to take a (substantial) toll on the outcome of the training in virtually every tested settings when half of the workers are Byzantine. We propose a simple method which carries no computational complexity and is yet able to recover at least 20% (sometimes 50% or even more is recovered, c.f. Section D, now Section E) in half of these settings, settings for which each of the 6 studied defenses have fallen.
> Such a gain may actually be considered in favor of the proposed method, at least when compared to its negligible cost.
>
> A second aspect is that this work is clearly leaned toward empirical assessments, but the review does not criticize them, and instead calls for more insights and analysis. A first, straightforward insight our experiments provide is that most of the studied defenses are actually ineffective against the two presented attacks in most of the 3680 tested settings.
>
> On the analytical side, our results highlights the fact that the 6 known state-of-the-art, theoretically proven, resilient gradient aggregation rules (GARs) are better analysed through the lens of the variance-ratio norm. Previous work (Baruch et al. NeurIPS 2019) have pointed empirical evidence for attacking respectively 3 and 2 of these rules (Krum, Median, Trimmed Mean), we provide an analysis spanning the 6 of them and providing new theoretical insights. These insights inform both on the theoretical assumptions of these GARs, but also provide a new defense mechanism we assess theoretically, and evaluate empirically on the known state-of-the-art attacks.
>
> On the empirical side, which is arguably this paper's most important part, we would like to stress the fact that our empirical evidence consists in 3680 runs, spanning a wide range of variation of hyperparameters* on the known state-of-the-art attacks, not on a few particular choice of hyperparameters, which increases the confidence one should put in this assessment of our method. We are also adding new experiments, with a much larger model, and the empirical observations remain the same: our method does improve the resilience of existing defenses. Our code enables any member of the community to assess this claim.
> *we cross-tested every possible combination (and each combination five times, with specific seeds) of the following hyperparameters: which attack is used, which defense (Krum, Median, Trimmed Mean, Phocas, MeaMed, Bulyan), how many Byzantine workers (an half or a quarter), where momentum is computed (server or workers), which flavor of momentum is used (classical or Nesterov), which learning rate is used (larger or smaller).
>
> We understand that the results of Section D (now Section E) might look not compressed enough, we are afraid even aggregates such as median, average, etc, are not enough to grasp their multi-faceted aspect (e.g. recovering 20% accuracy for a model that only reaches 40% accuracy is arguably not comparable to recovering 20% for a model that can reach above 90% accuracy).
> We added graphs in the same nature of Figure 1 for all the experiments, which we believe is a more accurate way of displaying aggregated accuracy gains/recoveries, as each graph cross-compares the 6 defenses against the 2 attacks using the same set of hyperparameters (same model, same dataset, same number of Byzantine workers, etc).
>
> [1] Ilya Sutskever, James Martens, George Dahl, and Geoffrey Hinton.
> On the importance of initialization and momentum in deep learning.
> ICML 2013.

---

### Official Review · AnonReviewer4 · 2020-10-29
**Extensive experiments with some theoretical analysis**

**Rating:** 7
**Confidence:** 4

**Review:**

# Contributions

This paper presents a novel method to tackle the Byzantine faults problem. By using a local momentum, this method can be extended to all other existing robust algorithms. The authors also provide some theoretical analysis of the effect of their algorithm. Finally, comprehensive experiments are shown and analyzed.

# Strong points

1. The local momentum can be easily combined with other existing robust algorithms, which makes it more practical.

3. This paper has comprehensive experiments that compare the combination of different attacks, defenses and datasets.

4. The authors provide well-written code to reproduce all experiments.

# Weak points

1. The experiments can be improved by running on larger datasets and larger models. Higher dimensionality may also impact the performance. But it is unrealistic to run thousands of large scale experiment, so this weak point is understandable.

2. The assumption is too strong. Namely, each worker can sample from the global dataset, and the norm of "real" gradients are bounded.

3. In the theoretical analysis, the variance-norm ration is defined as $r_t^{(s)} = \frac{\mathbb {E} \| \mathcal{G} - \mathbb {E} \mathcal{G} \|^2}{\| \mathbb {E} \mathcal{G} \|^2}$. However, when at the (local) optimal, $\lambda_t^2 = \| \mathbb {E} \mathcal{G} \|^2 = \mathbf 0$. Maybe define $r_t^{s}$ as $r_t^{(s)} = \frac{\| \mathbb {E} \mathcal{G} \|^2}{\mathbb {E} \| \mathcal{G} - \mathbb {E} \mathcal{G} \|^2}$ can resolve this problem and remove the requirement that $\lambda_t > 0$.

4. When stating the definition of $\nabla Q$ being Lipschitz, I think it should be $\| \mathbb {E} \mathcal{G}_t - \mathbb {E} \mathcal{G}_u \| \leq l^2 \| \theta_t - \theta_u \|^2$. But it doesn't affect the final result.

# Recommendation
Accept. This paper proposes a new way to solve the Byzantine faults problem, along with some theoretical analysis, extensive experiments and well-written code.

# Optional improvements

1. Page 3, the first sentence of Adversarial Model paragraph, "... as the minimization of ...", do you mean maximization?

2. Definition 1, $(\alpha, f) \in [0, .... \frac{\pi}{2} [ \times [0, ..., n]$, is it a typo?

# Update
Though the theoretical analysis is a bit weak, I think the experiments are quite good. The code can also run without any issue, which is a significant contribution in my opinion.

---

> ### Author Response · Authors · 2020-11-17
> **Response: Extensive experiments with some theoretical analysis**
>
> We thank the reviewer for their detailed review.
>
> We understand this concern, and we confirm it quickly becomes unrealistic to run such a large scale experiment with larger models. Specifically, we tried the "Wide_ResNet" from `https://github.com/meliketoy/wide-resnet.pytorch` (our code supports external models by just symlinking the appropriate Python module in `experiments/models/`) and the execution time is clearly prohibitive for 3680 runs: around 6h per run per GPU.
> Since we understand this concern, we are nevertheless running a subset of the experiments (namely: only Nesterov momentum and only CIFAR-10) with this pair model/dataset and the less memory-hungry GARs (we are trying to run all the GARs, but some have triggered _OOM errors_ on our GPUs; we hope these were only transient errors, so we'll retry them later).
> We have already run 28 different settings with seed 1 (we modified the run script to ran it first), but the 4 other seeds will take more time to complete. We have added in the paper new graphs for this pair model/dataset with all the data we already have with seed 1, and we will complete them once the remaining experiments finish (these pending experiments will make possible to compute and display standard deviations).
>
> The assumption that each worker samples unbiased estimates from the same distribution is arguably strong, at least for some practical deployments (e.g. federated learning). Such a strong requirement is inherited from the theoretical requirements of existing statistically-robust defenses, which we augment with our method.
> (Actually, it appears another team proposed a method to tackle this very issue, based on "resampling" gradients at the server (paper 450). After a quick reading, it also seems that our method would work with theirs: the resampled gradients are arithmetic means of submitted gradients, another linear operation which would then commute with the momentum computation at the workers.)
>
> Defining $r_t^{(s)}$ and $r_t^{(w)}$ as the norm-variance ratio would exchange the requirements in our mathematical developments, from requiring the norm of the (momentum) gradient to be strictly positive to requiring the (momentum) variance to be strictly positive. We appreciate the remark, that makes sense since the theoretical goal in non-convex SGD is to reach a local minimum, at which the real gradient is indeed zero, and one can most reasonably expect the honest stochastic gradient to have non-zero variance. Nevertheless, we believe you would agree such changes do not affect the essence of the analysis, and for mathematical completeness the case $\lambda_t = 0$ could be dealt with apart.
> A more delicate case (perhaps you meant this one) would happen if $\Lambda_t$ comes close to $0$, and we propose to tackle this scenario (as future work) by having each honest worker $i$ send $G_t^{(i)}$ when Equation 7 is satisfied, and $g_t^{(i)}$ otherwise.
> (While looking into your comment, we added missing requirements of non-zero variance for Equation 7.)
>
> Indeed, a square operation was missing for one of the norms. It has been corrected in the updated version.
>
> Regarding the adversarial model paragraph, we are talking from the perspective of the adversary: its goal is to prevent the training from improving the model accuracy, hence "minimizing" the accuracy instead of maximizing it.
>
> Perhaps the notation we used was non-standard. We changed it in the updated version by $0 \le \alpha < \frac{\pi}{2}$ and $f \in \left[ 0 .. n \right]$.

---

### Official Review · AnonReviewer1 · 2020-11-03
**Technical contribution seems to be lacking**

**Rating:** 4
**Confidence:** 3

**Review:**

I am having trouble identifying the contribution of this paper. A slight distributed tweak on the completely standard approach of running SGD with momentum is proposed as a defense to Byzantine attacks. A number of defenses augmented with this approach is studied experimentally against two recent attacks. There are no clear guarantees of the proposed mechanism that I could find as the paper seems to be focused exclusively on experimental results.

The experimental results by themselves, while they have a certain merit, seem of little lasting value -- there is only two (rather ad hoc) attacks considered, which I didn't find particularly natural or important otherwise. Overall, combined with the fact the Byzantine setting itself is already a stretch in terms of realism, this paper doesn't seem to have much lasting value. I would suggest identifying and studying broader classes of attacks/defenses of which presented ones are special cases and giving at least some guarantees in terms of what the proposed approach provides.

---

> ### Author Response · Authors · 2020-11-17
> **Response: Technical contribution seems to be lacking**
>
> We thank the reviewer for their review.
>
> The proposed idea is indeed simple, both to understand and to implement, which we actually view as a strength. Our second contribution is indeed to empirically study the effect of this simple idea, running over a large set of hyper-parameters, confirming that such a simple "distributed tweak" can actually improve the resilience of existing defenses; and sometimes substantially.
> Anticipating such a positive effect for such a simple change, especially since this change does not carry any computational complexity, does not seem obvious and could thus be overlooked by the community (building and running reproducible experiments to assess an idea which sounds like "free lunch" may not be something practitioners would be willing to spend time on).
>
> The studied attacks (Baruch et al., 2019; Xie et al., 2019a) are, to the best of our knowledge, the only two effective attacks available in the literature. We are willing to implement and test more attacks, and include them in the paper (there may be enough time to both write the code and get at least some of the results by the 24th), if pointers to relevant publications can be provided. Please note that we also tested but discarded the attack from (El-Mhamdi et al., 2018), since it does not have an impact as strong as the two others (the implementation is available in attacks/identical.py, line 89).
>
> The Byzantine setting is the broadest theoretical framework to study faults, and so it already encompasses all classes of attack. Regarding the defenses, we do not propose per-se a new defense but a method to improve the effectiveness of any statistically-robust defense. Applying our method to other classes of defenses, e.g. defenses based on redundancy schemes, may not be possible and is outside the scope of this paper.
>
> Reducing the variance-norm ratio for statistically-robust defenses is technically the purpose of our method, so we provide in the appendix an analysis of the impact of our method on the variance-norm ratio (we understand the appendix is optional to read).

---

### Decision · Program_Chairs · 2021-01-07
**Final Decision**

**Decision:**

Accept (Poster)

**Comment:**

The authors present a simple modification of existing byzantine resistant techniques for training in the presence of worst case failures/attacks. The paper studies two of the strongest attacks to date, that no other method, till now, has been able to address. The novelty is significant for the related byzantine ML literature. The authors further do a fantastic job in their experiments and sharing reproducible code. Some weak aspects of theory are in fact attributed to what the metrics and guarantees that the related literature studies. The novelty of this paper does not lie so much in the theory contribution, but more so on their experiments and presented intuition. I believe this will be a paper that people will build up on and the ideas presented here are of solid value and importane.